



# DCMIP2016: the tropical cyclone test case

Justin L. Willson[1], Kevin A. Reed[1], Christiane Jablonowski[2], James Kent[3], Peter H. Lauritzen[4],
Ramachandran Nair[4], Mark A. Taylor[5], Paul A. Ullrich[6], Colin M. Zarzycki[7], David M. Hall[8,9],
Don Dazlich[10], Ross Heikes[10], Celal Konor[10], David Randall[10], Thomas Dubos[11], Yann Meurdesoif[11],
Xi Chen[12,13], Lucas Harris[12], Christian Kühnlein[14], Vivian Lee[15], Abdessamad Qaddouri[15],
Claude Girard[15], Marco Giorgetta[16], Daniel Reinert[17], Hiroaki Miura[18], Tomoki Ohno[19], and
Ryuji Yoshida[20]

[1]School of Marine and Atmospheric Sciences, Stony Brook University, Stony Brook, NY, USA
[2]Department of Climate and Space Sciences and Engineering, University of Michigan, Ann Arbor, MI, USA
[3]School of Computing and Mathematics, University of South Wales, Pontypridd, Wales, UK
[4]National Center for Atmospheric Research, Boulder, CO, USA
[5]Sandia National Laboratories, Albuquerque, NM, USA
[6]Department of Land, Air and Water Resources, University of California, Davis, Davis, CA, USA
[7]Department of Meteorology and Atmospheric Science, Penn State University, University Park, PA, USA
[8]Department of Computer Science, University of Colorado, Boulder, Boulder, CO, USA
[9]NVIDIA Corporation, Santa Clara, CA, USA
[10]Department of Atmospheric Science, Colorado State University, Fort Collins, CO, USA
[11]IPSL/Lab. de Météorologie Dynamique, École Polytechnique, Palaiseau, France
[12]Geophysical Fluid Dynamics Laboratory (GFDL), National Oceanic and Atmospheric Administration, Princeton, NJ, USA
[13]Institute of Atmospheric Physics, Chinese Academy of Sciences, Beijing, China
[14]European Centre for Medium-Range Weather Forecasts (ECMWF), Bonn, Germany
[15]Environment and Climate Change Canada (ECCC), Dorval, Québec, Canada
[16]Department of the Atmosphere in the Earth System, Max Planck Institute for Meteorology, Hamburg, Germany
[17]Deutscher Wetterdienst (DWD), Offenbach am Main, Germany
[18]Department of Earth and Planetary Science, Graduate School of Science, The University of Tokyo, Tokyo, Japan
[19]Atmosphere and Ocean Research Institute, The University of Tokyo, Kashiwa, Japan.
[20]Division of Natural Environment and Information, The Yokohama National University, Kanagawa, Japan

**Correspondence:** Kevin A. Reed (kevin.reed@stonybrook.edu)

**Abstract.** This paper describes and analyzes the Reed-Jablonowski (RJ) tropical cyclone (TC) test case used in the 2016
Dynamical Core Model Intercomparison Project (DCMIP2016). The intermediate complexity test case analyzes the evolution
of a weak vortex into a TC in an idealized tropical environment. Simulations from 9 general circulation models (GCMs) that
participated in DCMIP2016 are analyzed in this study at 50 km horizontal grid spacing, with 5 of these models also providing

simulations at 25 km grid spacing for an analysis on the impact of finer grid spacing. Evolution of minimum surface pressure
(MSP) and maximum 1 km azimuthally averaged wind speed (MWS), the wind-pressure relationship, radial profiles of wind
speed and surface pressure, and wind composites are documented for all participating GCMs at both horizontal grid spacings.
While results are generally similar between all models, some GCMs reach significantly higher storm intensities than others,
ultimately impacting specific characteristics of their horizontal and vertical structure. TCs simulated at 25 km grid spacings

retained these differences, but reach higher intensities and are more compact than their 50 km counterparts. These results





indicate dynamical core choice is an essential factor in GCM development, and future work should be conducted to explore how specific differences within the dynamical core affect TC behavior in GCMs.

## 1 Introduction

Tropical cyclones (TCs) are among the most dangerous meteorological phenomena in the world, causing billions of dollars in
damage annually and significantly impacting both coastal and offshore regions (Emanuel, 2003). TC behavior is expected to change with global warming, with the most confident projection being increased storm surge and flooding due to higher sea levels (Knutson et al., 2020). There is also medium-to-high confidence that the global average intensity and precipitation rates of these storms will increase, and the proportion of TCs reaching a high intensity (categories 4 and 5 on the Saffir-Simpson scale) is expected to increase as well (Knutson et al., 2020). It is possible that anthropogenic signals in past observational data
have already been observed in some cases, such as the increase in the proportion of category 4 and 5 TCs in recent data and an increase in the global average intensity of the strongest TCs since the beginning of the 1980s (Knutson et al., 2019). Since TCs pose a significant risk to society, a risk that will likely increase in the future, it is important to understand them. Simulating TCs accurately in general circulation models (GCMs) accomplishes this through a better understanding of TC dynamics, changes in their future behavior, and how they interact with other processes on climate scales.

Simulating TCs in GCMs is complicated and requires both extensive computational power and sophisticated simulations. Although the typical CMIP-class GCM grid spacing is around 100 km, TCs are not well resolved when the grid spacing is coarser than 50km. This is due to the small size of TCs and the sophisticated physical processes that cause their formation and propagation (Reed and Jablonowski, 2011a). Simulated TC characteristics in GCMs such as intensity, size, and genesis generally become more accurate with increasing horizontal resolution (Reed and Jablonowski, 2011a, c), a trend that has also
been shown in decadal, climate-scale simulations (Wehner et al., 2014; Stansfield et al., 2020; Roberts et al., 2020). Reed and Jablonowski (2011a) describes the initialization of an initially weak warm-core idealized vortex into a TC during a 10 day period in a GCM with an environment conducive for storm development. The simulated TC increases in intensity and becomes more compact with increasing horizontal resolution (Reed and Jablonowski, 2011a). The TC test case developed in Reed and Jablonowski (2011a), hereafter referred to as the Reed-Jablonowski (RJ) TC test case, is a moist deterministic TC test case of
intermediate complexity designed to be used in simple physics experiments with complexities between dry dynamical core and full physics aquaplanet simulations. This test case provides a less complex regime to study how the dynamical core and moist physical parameterizations interact without having to conduct a computationally expensive simulation (Reed and Jablonowski, 2012).

There are several studies that demonstrate the usefulness of studying the RJ TC test case formulation. Reed and Jablonowski
(2011b) found that the RJ TC test case increases in strength and size while having an earlier onset of intensification in the Community Atmosphere Model (CAM) version 4 (CAM4), developed by the National Center for Atmospheric Research (NCAR), compared to its predecessor CAM3 due to the presence of a dilute plume convective available potential energy (CAPE) calculation in CAM4. Additionally, Reed and Jablonowski (2011c) illustrates how uncertainties in the RJ TC test case simulations





can be structural, parameter based, or initial-data based, with structural uncertainties being the most prominent when CAM4

and CAM5 were compared. This study did not take into account structural differences in the dynamical core of the models, the component that integrates the Navier-Stokes equations that describe fluid motion in the atmosphere, likely underestimating structural uncertainty (Reed and Jablonowski, 2011c). Reed and Jablonowski (2012) investigated how the dynamical core choice impacts the RJ TC test case structure and strength in simple physics simulations and complex CAM5 full-physics aquaplanet simulations with grid spacings of approximately 50 km or less. This work indicated that simple physics experiments

can provide meaningful insight into how dynamical core characteristics, including resolution, impact simulated TC behavior. In CAM5 comprehensive climate-scale simulations, the spectral element dynamical core produces more TCs that tend to have stronger intensities than those produced in a simulation using the finite volume dynamical core (Reed et al., 2015). This result was consistent with the RJ TC test case analysis in Reed and Jablonowski (2012), demonstrating further confidence for the use of idealized test cases and intermediate complexity simulations for better understanding the impact of GCM design on

simulated TC characteristics.

Other GCM characteristics have been shown to have a significant impact on the resulting behavior of the RJ TC test case. The grid spacing of the model is critical to simulating certain characteristics in the RJ TC test case, with higher resolution models creating increasingly intense and compact TCs that can even demonstrate non-physical intensity due to the physics parameterization behavior at small time steps (Reed et al., 2012). He et al. (2018) uses Sobol's variance-based sensitivity anal-

ysis to analyze input/output relationships that are multivariate in nature, and demonstrates that resolution significantly impacts sensitivities to control factors, with coarse resolution simulations unable to produce an accurate TC. This study also found non-linear relationships between factors that control precipitation rate, cloud content, and radiative forcing in the idealized RJ TC test case described previously. He and Posselt (2015) demonstrates how the parameterized physical processes in cloud formation, convective development, and moist turbulence impact the simulation of TC intensity, precipitation rate, and other

characteristics during the evolution of this RJ TC test case in CAM5, with none of the model output variables having a linear response. In CAM5, the precipitation and intensity of the simulated TCs are sensitive to the physics-dynamics timestep, with the magnitude of the sensitivities dependent on the dynamical core and horizontal resolution used (Li et al., 2020).

The 2016 Dynamical Core Model Intercomparison Project (DCMIP2016) described in Ullrich et al. (2017) aims to increase knowledge about how the dynamical core of the climate model impacts the resulting behavior of various meteorological test

cases. The test cases used in DCMIP2016 (Ullrich et al., 2016) include simplified moist physics and build upon the previous sets of test cases developed for DCMIP2012 (Ullrich et al., 2012) and DCMIP2008 (Jablonowski et al., 2008). These test cases include a moist baroclinic wave, a splitting supercell (Zarzycki et al., 2019), and the RJ TC test case, which will be documented in this study. More information about the RJ TC test case used will be given in Section 2.1. Climate models, especially high resolution GCMs, are computationally expensive and many model development groups have found computational methods and

algorithms to reduce this complexity; therefore, there are many formulations of the dynamical core that exist. While studies have explored such processes in the past, they largely focus on the effects of changing certain parameters in one model. This model intercomparison study allows extensive comparison and documentation of TC behavior among different GCMs, which have differences in their dynamical core design such as the horizontal and vertical discretization, time stepping, native grid,





and grid staggering (Ullrich et al., 2017). Detailed reference solutions to prescribed test cases among models with varying

characteristics contribute to the improvement of GCMs and the simulation of TCs. This study provides an intercomparison of RJ TC test case behavior among the DCMIP2016 models. The goal of this analysis is to provide a library of solutions that serve as a benchmark for modeling groups to compare against and does not aim to link differences in results to specific numerical differences in the models. The RJ TC test case is in wide use among modeling groups, and many of these groups regularly utilize DCMIP test cases for internal use. Further, some DCMIP test cases are members of the Community Earth System Model

(CESM) Simpler Models framework, indicating their importance to the community (CES).

Section 2 provides a detailed explanation of the initialization of the RJ TC test case along with the analytical and computational procedures used throughout this study. Section 3 catalogs similarities and differences of the RJ TC test case behavior among DCMIP2016. Specifically, we analyze the evolution of maximum 1 km azimuthally averaged wind speed (MWS) and minimum surface pressure (MSP), radial profiles and vertical composites of wind speed and surface pressure, and the wind-

pressure relationship in the TC. Differences between the 50 km and the 25 km simulations are also discussed for models that submitted simulations at both grid spacings. Section 4 summarizes important results from the model intercomparison and provides a motivation for future work in this area.

## 2  Methods

### 2.1  Description of Tropical Cyclone Test Case

The idealized RJ TC test case used here is based on the work of Reed and Jablonowski (2011a, 2012). A weak balanced vortex is initialized in an environment conducive to rapid intensification, and this initial vortex evolves into a TC over a period of 10 days. One of the goals of the RJ TC test case is to explore the impact of physics-dynamics coupling and grid resolution sensitivities on TC simulation in GCMs. In DCMIP2016, the RJ TC test case is one of 3 test cases, and it was run for each of the models using a simplified physical parameterization package in a controlled testing environment to allow for the analysis of

physics-dynamics interactions and other characteristics (Ullrich et al., 2016). For reproducibility, lists of DCMIP2016 model initialization symbols, physical constants, and RJ TC test case constants are shown in Table 1, Table 2, and Table 3 respectively. The complete mathematical description of the initialization and axisymmetric vortex of the RJ TC test case is in the subsequent sections.

### 2.1.1  Environmental Background

The RJ TC test case is initialized in the following manner. It contains a background state that consists of three profiles: prescribed specific humidity, virtual temperature, and pressure. The initial profile is in an approximate gradient wind balance state by definition. The vertical sounding is chosen to approximately resemble an observed tropical sounding (Jordan, 1958).





**Table 1.** List of DCMIP2016 symbols used in model initialization (Ullrich et al., 2016).

| Symbol | Description |
| --- | --- |
| $\lambda$ | Longitude (in radians) |
| $\varphi$ | Latitude (in radians) |
| $z$ | Height with respect to mean sea level (set to zero) |
| $p_s$ | Surface pressure ($p_s$ of moist air if $q > 0$) |
| $\Phi_s$ | Surface geopotential |
| $z_s$ | Surface elevation with respect to mean sea level (set to zero) |
| $u$ | Zonal wind |
| $v$ | Meridional wind |
| $w$ | Vertical velocity |
| $\omega$ | Vertical pressure velocity |
| $\delta$ | Divergence |
| $\zeta$ | Relative vorticity |
| $p$ | Pressure (pressure of moist air if $q > 0$) |
| $\rho$ | Density (density of moist air if $q > 0$) |
| $T$ | Temperature |
| $T_v$ | Virtual temperature |
| $\Theta$ | Potential temperature |
| $\Theta_v$ | Virtual potential temperature |
| $q$ | Specific humidity |
| $P_{ls}$ | Large-scale precipitation rate |
| $q_c$ | Cloud water mixing ratio |
| $q_r$ | Rain water mixing ratio |
| $q_{Cl}$ | Singlet chlorine mixing ratio |
| $q_{Cl2}$ | Chlorine gas mixing ratio |





**Table 2.** A list of physical constants used in DCMIP2016 (Ullrich et al., 2016).

| Constant | Description | Value |
|---|---|---|
| $a_{ref}$ | Radius of the Earth | $6.37122 \times 10^6$ m |
| $\Omega_{ref}$ | Rotational speed of the Earth | $7.292 \times 10^{-5}$ s$^{-1}$ |
| $X$ | Reduced-size Earth reduction factor | variable (default $= 1$) |
| $a$ | Scaled radius of the Earth | $a_{ref}/X$ |
| $\Omega$ | Scaled rotational speed of the Earth | $\Omega_{ref} \cdot X$ |
| $g$ | Gravity | $9.80616$ m s$^{-2}$ |
| $p_0$ | Reference pressure | 1000 hPa |
| $c_p$ | Specific heat capacity of dry air at constant pressure | 1004.5 J kg$^{-1}$ K$^{-1}$ |
| $c_v$ | Specific heat capacity of dry air at constant volume | 717.5 J kg$^{-1}$ K$^{-1}$ |
| $R_d$ | Gas constant for dry air | 287.0 J kg$^{-1}$ K$^{-1}$ |
| $R_\nu$ | Gas constant for water vapor | 461.5 J kg$^{-1}$ K$^{-1}$ |
| $\kappa$ | Ratio of $R_d$ to $c_p$ | 2/7 |
| $\varepsilon$ | Ratio of $R_d$ to $R_\nu$ | 0.622 |
| $M_v$ | Constant for virtual temperature conversion | 0.608 |
| $\rho_{water}$ | Density of water | 1000 kg m$^{-3}$ |

**Table 3.** List of constants used for the Idealized Tropical Cyclone test (Ullrich et al., 2016).

| Constant | Value | Description |
|---|---|---|
| $X$ | 1 | Small-planet scaling factor (regular-size Earth) |
| $z_t$ | 15000 m | Tropopause height |
| $q_0$ | 0.021 kg/kg | Maximum specific humidity amplitude |
| $q_t$ | $10^{-11}$ kg/kg | Specific humidity in the upper atmosphere |
| $T_0$ | 302.15 K | Surface temperature of the air |
| $T_s$ | 302.15 K | Sea surface temperature (SST), 29 C$^\circ$ |
| $z_{q1}$ | 3000 m | Height related to the linear decrease of $q$ with height |
| $z_{q2}$ | 8000 m | Height related to the quadratic decrease of $q$ with height |
| $\Gamma$ | 0.007 K m$^{-1}$ | Virtual temperature lapse rate |
| $p_b$ | 1015 hPa | Background surface pressure |
| $\varphi_c$ | $\pi/18$ | Initial latitude of vortex center (radians) |
| $\lambda_c$ | $\pi$ | Initial longitude of vortex center (radians) |
| $\Delta p$ | 11.15 hPa | Pressure perturbation at vortex center |
| $r_p$ | 282000 m | Horizontal half-width of pressure perturbation |
| $z_p$ | 7000 m | Height related to the vertical decay rate of $p$ perturbation |
| $\epsilon$ | $10^{-25}$ | Small threshold value |



The background specific humidity profile $\overline{q}(z)$ as a function of height $z$ is

$$\overline{q}(z) = q_0 \exp\left(-\frac{z}{z_{q1}}\right) \exp\left[-\left(\frac{z}{z_{q2}}\right)^2\right] \quad \text{for } 0 \leq z \leq z_t$$

$$\overline{q}(z) = q_t \quad \text{for } z_t \leq z \tag{1}$$

The specific form of the background virtual temperature sounding $\overline{T}_v(z)$ is dependent on its location in the atmosphere. In this case, there are two different representations, one for the lower atmosphere and another for the upper atmosphere. It is given by

$$\overline{T}_v(z) = T_{v0} - \Gamma z \qquad \text{for } 0 \leq z \leq z_t,$$
$$\overline{T}_v(z) = T_{vt} = T_{v0} - \Gamma z_t \quad \text{for } z_t < z, \tag{2}$$

with the virtual temperature at the surface $T_{v0} = T_0(1 + 0.608 q_0)$ and the virtual temperature at the tropopause level $T_{vt} = T_{v0} - \Gamma z_t$. The background temperature profile can be obtained from the equation:

$$T_v = T(1 + M_v q) \tag{3}$$

The background vertical pressure profile $\overline{p}(z)$ of the moist air can be calculated using the hydrostatic balance and (2). The profile is given by:

$$\overline{p}(z) = p_b \left(\frac{T_{v0} - \Gamma z}{T_{v0}}\right)^{g/R_d\Gamma} \quad \text{for } 0 \leq z \leq z_t,$$
$$\overline{p}(z) = p_t \exp\left(\frac{g(z_t - z)}{R_d T_{vt}}\right) \quad \text{for } z_t < z. \tag{4}$$

The pressure at the tropopause level $z_t$ is continuous and given by

$$p_t = p_b \left(\frac{T_{vt}}{T_{v0}}\right)^{\frac{g}{R_d\Gamma}}, \tag{5}$$

This value is approximately 130.5 hPa for the set of parameters used in the test case initialization.

### 2.1.2 Axisymmetric Vortex

The pressure equation $p(r, z)$ for the moist air is composed of the background pressure profile (4) and a 2D pressure perturbation $p'(r, z)$,

$$p(r, z) = \overline{p}(z) + p'(r, z), \tag{6}$$

where $r$ symbolizes the radial distance (or radius) to the center of the prescribed vortex. On the sphere $r$ is defined using the great circle distance (GCD)

$$r = a \arccos\left(\sin\varphi_c \sin\varphi + \cos\varphi_c \cos\varphi \cos(\lambda - \lambda_c)\right). \tag{7}$$





The perturbation pressure is defined as

$$p'(r,z) = -\Delta p \exp\left[ -\left(\frac{r}{r_p}\right)^{3/2} - \left(\frac{z}{z_p}\right)^2 \right] \left(\frac{T_{v0} - \Gamma z}{T_{v0}}\right)^{\frac{g}{R_d \Gamma}} \qquad \text{for } 0 \le z \le z_t,$$

$$p'(r,z) = 0 \qquad \text{for } z_t < z. \qquad (8)$$

There are multiple pressure differences that impact the pressure perturbation. These include the pressure difference $\Delta p$ between the background surface pressure $p_b$ and the pressure at the center of the initial vortex, the pressure change in the radial direction $r_p$ and the pressure decay with height within the vortex $z_p$. The moist surface pressure $p_s(r)$ is computed by setting $z = 0$ m
in (6), which gives

$$p_s(r) = p_b - \Delta p \exp\left[ -\left(\frac{r}{r_p}\right)^{3/2} \right]. \qquad (9)$$

The axisymmetric virtual temperature $T_v(r,z)$ is calculated using the hydrostatic equation and ideal gas law

$$T_v(r,z) = -\frac{g p(r,z)}{R_d} \left(\frac{\partial p(r,z)}{\partial z}\right)^{-1}. \qquad (10)$$

Again this equation takes the form of a sum of the background state and a perturbation,

$$T_v(r,z) = \overline{T}_v(z) + T_v'(r,z), \qquad (11)$$

where the virtual temperature perturbation is defined as

$$T_v'(r,z) = (T_{v0} - \Gamma z)\left\{ \left[ 1 + \frac{2 R_d (T_{v0} - \Gamma z) z}{g z_p^2 \left[ 1 - \frac{p_b}{\Delta p} \exp\left( \left(\frac{r}{r_p}\right)^{3/2} + \left(\frac{z}{z_p}\right)^2 \right) \right]} \right]^{-1} - 1 \right\} \qquad \text{for } 0 \le z \le z_t,$$

$$T_v'(r,z) = 0 \qquad \text{for } z_t < z. \qquad (12)$$

The axisymmetric specific humidity $q(r,z)$ is set to the background profile everywhere

$$q(r,z) = \overline{q}(z). \qquad (13)$$

Consequently, the temperature can be written as

$$T(r,z) = \overline{T}(z) + T'(r,z), \qquad (14)$$

with the temperature perturbation

$$T'(r,z) = \frac{T_{v0} - \Gamma z}{1 + 0.608 \overline{q}(z)} \left\{ \left[ 1 + \frac{2 R_d (T_{v0} - \Gamma z) z}{g z_p^2 \left[ 1 - \frac{p_b}{\Delta p} \exp\left( \left(\frac{r}{r_p}\right)^{3/2} + \left(\frac{z}{z_p}\right)^2 \right) \right]} \right]^{-1} - 1 \right\} \qquad \text{for } 0 \le z \le z_t,$$

$$T'(r,z) = 0 \qquad \text{for } z_t < z. \qquad (15)$$





The upper troposphere has a small specific humidity value ($10^{-11}$ kg/kg for $z > z_t$); therefore, the virtual temperature approximately equals the temperature for this portion of the atmosphere. The formulation introduced here is equivalent to the one explained in Reed and Jablonowski (2012).

In some cases, the density of the moist air needs to be initialized as well. The ideal gas law forms the basis of this initialization and the density of the moist air is initialized in the following manner

$$\rho(r,z) = \frac{p(r,z)}{R_d T_v(r,z)} \tag{16}$$

which utilizes the moist pressure (6) and virtual temperature (11). The surface elevation $z_s$ and thereby the surface geopotential $\Phi_s = gz_s$ are set to zero.

Finally, gradient-wind balance, a function of pressure (6) and the virtual temperature (12), allows for the definition of the tangential velocity field $v_T(r,z)$ of the axisymmetric vortex. The tangential velocity is given by

$$v_T(r,z) = -\frac{f_c r}{2} + \sqrt{\frac{f_c^2 r^2}{4} + \frac{R_d T_v(r,z)\, r}{p(r,z)} \frac{\partial p(r,z)}{\partial r}}, \tag{17}$$

where $f_c = 2\Omega \sin(\varphi_c)$ is the Coriolis parameter at the constant latitude $\varphi_c$. Substituting $T_v(r,z)$ and $p(r,z)$ into (17) gives

$$v_T(r,z) = -\frac{f_c r}{2} + \sqrt{\frac{f_c^2 r^2}{4} - \frac{\frac{3}{2}\left(\frac{r}{r_p}\right)^{3/2}(T_{v0} - \Gamma z)R_d}{1 + \frac{2R_d(T_{v0}-\Gamma z)z}{gz_p^2} - \frac{p_b}{\Delta p}\exp\left(\left(\frac{r}{r_p}\right)^{3/2} + \left(\frac{z}{z_p}\right)^2\right)}} \qquad \text{for } 0 \leq z \leq z_t,$$

$$v_T(r,z) = 0 \qquad\qquad\qquad\qquad \text{for } z_t < z. \tag{18}$$

The tangential velocity is then separated (18) into its zonal and meridional wind components $u(\lambda,\varphi,z)$ and $v(\lambda,\varphi,z)$. Similar to Nair and Jablonowski (2008) these are calculated in the following way,

$$d_1 = \sin\varphi_c \cos\varphi - \cos\varphi_c \sin\varphi \cos(\lambda - \lambda_c) \tag{19}$$

$$d_2 = \cos\varphi_c \sin(\lambda - \lambda_c) \tag{20}$$

$$d = \max\left(\epsilon, \sqrt{d_1^2 + d_2^2}\right), \tag{21}$$

which are utilized in the projections

$$u(\lambda,\varphi,z) = \frac{v_T(\lambda,\varphi,z)\, d_1}{d} \tag{22}$$

$$v(\lambda,\varphi,z) = \frac{v_T(\lambda,\varphi,z)\, d_2}{d}. \tag{23}$$

$\epsilon = 10^{-25}$ is utilized to avoid divisions by zero. In this case, the vertical velocity is set to zero.

## 2.2 Simulation Design

The RJ TC test case was simulated in 9 GCMs. Information about these models can be found in Table 4. Models submitted test case runs prior to 2016 and all results are for iterations of the models at that point in time. The GCMs have likely been



**Table 4.** Information about models that submitted RJ TC test case simulations in DCMIP2016 and were analyzed in this study.

| Abbreviation | Full Name | Modeling Center/Group |
|---|---|---|
| ACME-A | Energy Exascale Earth System Model | Sandia National Laboratories and University of Colorado, Boulder, USA |
| CAM-SE | Community Atmosphere Spectral Element Model | National Center for Atmospheric Research, USA |
| CSU | Colorado State University Model | Colorado State University, USA |
| DYNAMICO | DYNAMical core on the ICOsahedron | Institut Pierre Simon Laplace (IPSL), France |
| FV$^3$ | GFDL Finite-Volume Cubed-Sphere Dynamical Core | Geophysical Fluid Dynamics Laboratory, USA |
| FVM | Finite Volume Module of the Integrated Forecasting System | European Centre for Medium-Range Weather Forecasts |
| GEM | Global Environmental Multiscale model | Environment and Climate Change Canada, Canada |
| ICON | ICOsahedral Non-hydrostatic model | Max-Planck-Institut für Meteorologie, Germany |
| NICAM | Non-hydrostatic Icosahedral Atmospheric Model | AORI/JAMSTEC/AICS, Japan |

**Table 5.** Additional information about the models used in this study. Three dynamical cores are present: spectral element (SE), finite difference (FD), and finite volume (FV). More information can be found in Ullrich et al. (2017)

| Abbreviation | Native Grid | Horizontal Grid Spacing (km) | Dynamical Core | Hydrostatic |
|---|---|---|---|---|
| ACME-A | Cubed sphere | 50,25 | SE | No |
| CAM-SE | Cubed sphere | 50,25 | SE | Yes |
| CSU | Geodesic | 50 | FV | Yes |
| DYNAMICO | Geodesic | 50 | FV | Yes |
| FV$^3$ | Cubed sphere | 50 | FV | No |
| FVM | Octahedral | 50,25 | FV | No |
| GEM | Yin-Yang | 50,25 | FD | No |
| ICON | Icosahedral triangular | 50 | FV | No |
| NICAM | Geodesic | 50,25 | FV | No |

updated since 2016, but it is still important to provide a set of benchmark solutions for the modeling community. For all models, a simulation with a horizontal grid spacing of 50 km, the default in DCMIP2016, was analyzed. For select models, intercomparison also took place at a finer horizontal grid spacing of 25 km. The models that completed a 25 km simulation

180 are ACME-A, CAM-SE, FVM, GEM, and NICAM. The computational efficiency of each model, that is its total time to give a solution at a particular resolution, is not considered here but is nonetheless important since certain models can operate at a higher effective resolution for the same computational cost. Some models are hydrostatic while others were non-hydrostatic, although this is unlikely to have a significant effect on the simulation (Liu et al., 2022). Additional simulation information is summarized in Table 5.

185 Simulations were performed over a 10 day period with 30 vertical levels on an interpolated latitude-longitude grid. Models submitted identical simulation runs on their native grid given in Table 5 and an interpolated latitude-longitude grid with co-



located (Arakawa-A type) data and grid spacing comparable to that of the native grid model run (Ullrich et al., 2016). Since all models submitted interpolated latitude-longitude runs, this grid was used for analysis. The vertical levels were either pressure-based levels (pressure or hybrid vertical coordinates) or height levels, with the lowest height level being 60-70 m above the surface. Models that used a pressure-based vertical level system were converted to a height scale by first converting to pressures if the model utilized hybrid coordinates, then converting the pressure levels at each latitude and longitude point to height values using the hypsometric equation. The height levels and corresponding variables of all models were then interpolated to desired height values using linear interpolation. The model configuration is a full aqua-planet setup with prescribed sea surface temperatures (SSTs) set to a constant 302.15 K. This initialization follows the analytic framework described in Sections 2.1.1 and 2.1.2.

The simple physics package used in these simulations has several key features. First, the large-scale condensation does not incorporate a cloud stage; therefore, there is no carrying of any condensates and no re-evaporation at lower vertical levels as excess moisture is removed instantaneously. This configuration also allows all condensed water vapor to be removed as precipitation at the surface. Second, the surface fluxes determine the atmosphere-ocean interactions and eddy diffusivities in the boundary layer parameterization in the simulation. In total, the physics package describes four surface fluxes: zonal velocity, meridional velocity, temperature, and specific humidity. The planetary boundary layer is the third major component of the simple physics package. The planetary boundary layer is defined as all levels with a pressure greater than 850 hPa, which gives an approximate boundary layer depth of 1-1.5 km. Potential temperature is used for the boundary layer parameterization in this case since its vertical profile effectively indicates static stability. The boundary layer here represents Ekman-like profiles characterized by turbulent mixing with a constant vertical eddy diffusivity. These boundary layer diffusivities are simplified in nature as they ignore eddy diffusivity dependence on complicated static stability indicators and represent a first order coupling to the dynamic conditions. All processes in the simple physics package are coupled using the time-splitting method. This simple physics configuration is identical to the one described in Reed and Jablonowski (2012).

### 2.3 Analysis Approach

We utilized TempestExtremes (Ullrich et al., 2021) to identify and gain information about the TC in each model using the following TempestExtremes commands.

1. DetectNodes –in_data_list $input_files –out $detectnodes_output –closedcontourcmd "PS,200.0,5.5,0" –mergedist 6.0 –searchbymin PS –outputcmd "PS,min,0;_VECMAG(U,V),max,2"

2. StitchNodes –in_fmt "lon,lat,PS,wind" –range 8.0 –mintime 10 –maxgap 3 –in $detectnodes_output –out $stichnodes_output –out_file_format "csv" –threshold "wind,>=,10.0,10;lat,<=,50.0,10;lat,>=,-50.0,10"

The commands perform the following functions. DetectNodes first finds the MSP and keeps the candidate point if there is not a lower pressure within 6° GCD and there is a 200 Pa increase in surface pressure within 5.5° GCD of the minimum. The command also records the MSP and MWS value at that timestep. StitchNodes is used to stitch candidates throughout time

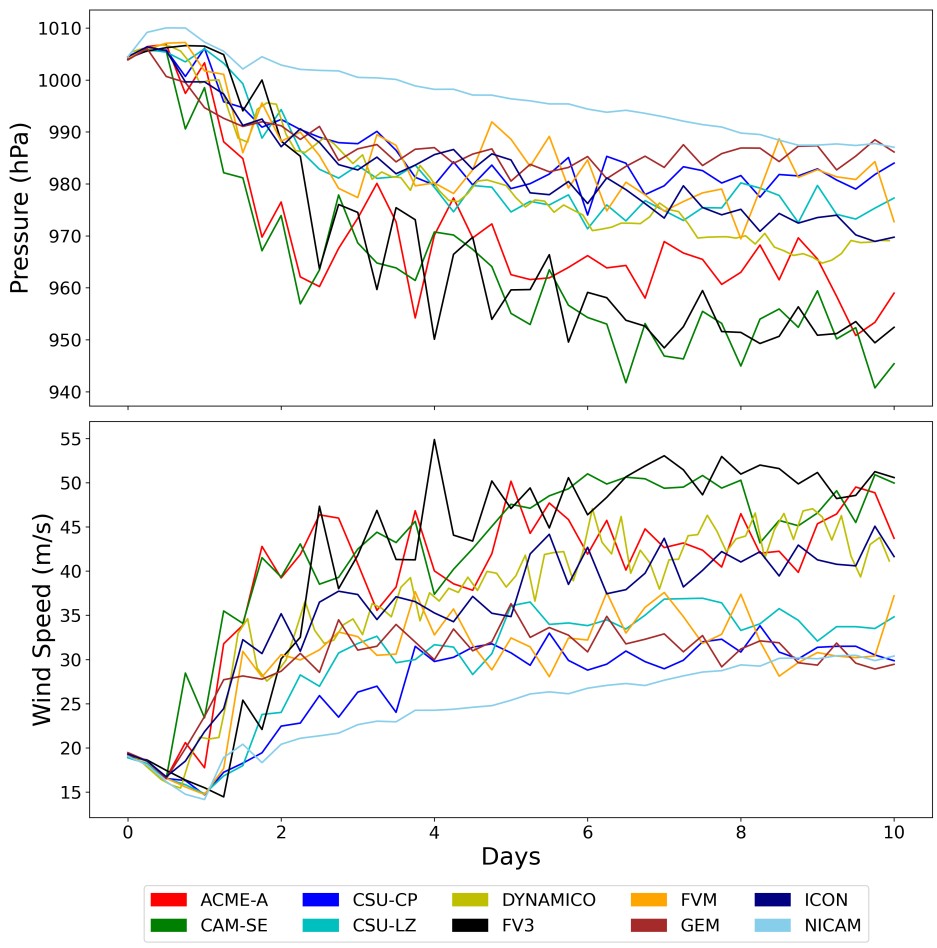

**Figure 1.** Evolution of MSP and MWS over the 10 day simulation period for the 50 km grid spacing.

into a trajectory. Candidate points are only stitched together if they are within $8°$ GCD at subsequent time steps, have a center

latitude magnitude less than $50°$, and have a lowermost model level wind speed greater than or equal to $10$ ms$^{-1}$.

From this analysis, the evolution of MWS and MSP along with the wind-pressure relationship of the TC were analyzed. Subsequently, radial profiles of 1 km wind speed and surface pressure were calculated using the following set of TempestExtremes commands:

1. NodeFileEditor –in_data_list \$input_files –in_nodefile \$stichnodes_output –out_nodefile \$wind_radprof_file

–out_nodefile_format "csv" –calculate "rprof=radial_wind_profile(U,V,159,0.25);rsize=lastwhere(rprof,>,8)" –out_fmt
     "lon,lat,rsize,rprof"

This command calculates the radial wind profile in the following manner. Using the StitchNodes output as input, the azimuthally averaged radial wind profile is obtained by first splitting the wind values into radial and azimuthal components, as





determined by the TC's center point, and then calculating the average based on the binning criteria. When calculating the sur-
face pressure radial profiles, the radial_wind_profile function was substituted for the radial_profile function. In this case, there
are 159 bins, each with a size of $0.25°$, in the radial profile. The resulting radial profile values were plotted at the midpoints of
these bins beginning around 14 km from the TC's center. Wind composites were also constructed using TempestExtremes to
identify similarities and differences in TC vertical structure between the models. This process was conducted by running the
same command for creating a radial wind profile at a series of height values. The wind composites are therefore azimuthally
averaged wind radial profiles at various heights, 0.1 km to 16 km for this analysis.

Both the radial profiles and the vertical wind composites were constructed during the semi-steady state period of the sim-
ulation, defined as the time when the TC MWS and precipitation rate were no longer undergoing a significant intensification
and any changes in their values were largely due to fluctuations within each model. The beginning of this steady state period
was estimated using a finite difference method combined with examination of the evolution of MWS and MSP. The evolution
of these quantities is shown in Figure 1. The rapid intensification of the TC during days 1-4 of the simulation period is clearly
seen in this figure, consistent with the RJ TC test case as previously analyzed (Reed and Jablonowski, 2011b, 2012). While
the intensity initially decreases, these trends reverse and the TC rapidly intensifies. This intensification process continues until
approximately day 4, when most DCMIP models appear to vary randomly around a fixed mean MWS or MSP. The beginning
of the steady state period was therefore determined to be day 4 of the simulations. Taking this into account, the radial profiles
and the vertical wind composites were averaged from days 4-10 of the simulation document the TC's structure during its stable
period with its maximum intensity.

The wind-pressure relationship within each TC was analyzed by plotting the corresponding MWS and MSP values at each
time step. A second-order polynomial function was then fit using a least squares method to each set of points to quantify this
relationship. This method has been used in several prior studies, including Reed et al. (2015) where it was used to quantify
the wind-pressure relationship in multiple CAM simulations and IBTrACS data, Knaff and Zehr (2007) where it was used to
fit observational aircraft pressure data and best-track wind data, and Kossin (2015) where it was used to fit the wind-pressure
relationship during eyewall replacement cycles seen in low-level aircraft reconnaissance data.

## 3  Results

The following section describes the results from the intercomparison, with the intent of highlighting differences in the resulting
TC behavior across the DCMIP ensemble. Although various model details were briefly mentioned in section 2.2 (and in more
detail in Ullrich et al., 2017), this analysis will not try to attribute individual model characteristics as the reason for the
differences in simulated TC behavior. Instead, we aim to provide an overview of the RJ TC test case results in DCMIP2016,
as well as discern characteristics of model groups based on similar TC behavior or highlight differences between one or more
models in certain areas, in order to provide a catalog of solutions that serve as a benchmark for modeling groups.





## 3.1 Time Evolution of Wind Speed and Pressure

The evolution of the MWS and the MSP is shown in Figure 1. The MSP initially increases in all models, a sign of weakening of the vortex which was also seen in simple physics simulations in Reed and Jablonowski (2012) as well as more complex full physics simulations in Reed and Jablonowski (2011b, 2012). All models then begin to intensify as the MSP decreases, and 3 model groups form shortly after day 2. NICAM retains the highest MSP, and consistently decreases in pressure throughout the simulation without much variation. CSU-CP, CSU-LZ, DYNAMICO, GEM, FVM, and ICON decrease to values in between 970 hPa and 990 hPa and vary within this range at subsequent time steps. The models increasingly diverge in the latter portion of the simulation, especially during days 7 to 10, and by day 9 NICAM enters this pressure range and becomes part of this model grouping. ACME-A, CAM-SE, and FV$^3$ continue decreasing until approximately day 4, 1-2 days later than the previous group of models, and then generally remain in the 950 hPa to 970 hPa range. These models all contain high variation in MSP changes compared to the other models, with differences in 5 hPa or above routinely seen at adjacent time steps. All MSP values are physically viable in the simulations, with none decreasing below 940 hPa.

Similarly, all models initially experience a decrease in their MWS during the first 1-2 days of the simulation. Then, all models undergo a rapid intensification until approximately day 4 after which they enter a steady state, behavior quantified in Section 2.3. There are exceptions to this trend; for example, NICAM continually increases throughout the simulation period with little variation. The models again split into groups, in this case after day 5 of the simulation. The models that occupy the lower MWS range are CSU-CP, CSU-LZ, FVM, GEM, and NICAM. After the intensification period, they generally have MWS values between 25 ms$^{-1}$ and 35 ms$^{-1}$. The second group of models, ACME-A, CAM-SE, FV$^3$, DYNAMICO, and ICON, have MWS values generally between 35 ms$^{-1}$ and 55 ms$^{-1}$. The groupings of models were slightly different than the evolution of MSP, with DYNAMICO and ICON being among the models that had increased MWS values despite the fact that they were among the models in the higher MSP range. In all models except NICAM, the variability was approximately equal, with the majority of MWS changes in adjacent timesteps being under 10 ms$^{-1}$ after day 4. MWS values were physically possible as their maximum magnitude corresponds to tropical storm to category 3 strength on the Saffir-Simpson scale.

## 3.2 Wind-Pressure Relationship

Figure 2 displays the MWS against the MSP at all time steps in the TC's evolution. In all models, MWS increases as MSP decreases, and this increase is nonlinear in most cases, especially at high intensities. As in the analysis of the evolution of MSP and MWS, there are groupings of the models that display similar wind-pressure relationships, and these groupings generally map to groupings seen in Figure 1. The wind-pressure relationships of all the models were physically possible, since the MSP and MWS were within the observed ranges of these variables. These ranges are seen in plots of observational data in Knaff and Zehr (2007) and Reed et al. (2015), which show MSP has an approximate range of 870-1015 hPa and MWS has an approximate range of 8-85 ms$^{-1}$. The models with the largest MWS values and the smallest MSP values in Figure 1 tended to have the largest range of MSP and MWS since all models initially start at the same surface pressure and wind speed.

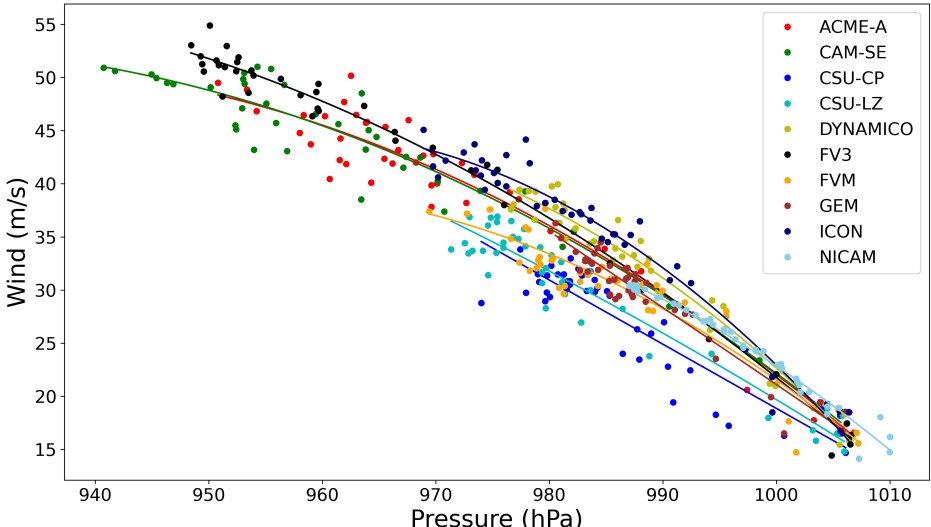

**Figure 2.** Wind-pressure relationship in the simulated TC at all time steps for the 50 km grid spacing. MWS and MSP from Figure 1 were used in this calculation. Second-order polynomial functions are fit using a least squares method.

ACME-A, CAM-SE, and FV[3] all display these characteristics. They have relatively low numbers of occurrences in the areas of large MSP and small MWS, indicating again how they quickly intensify in the first 1-2 days of the simulation with low variability, the range of MWS seen at a particular MSP. The majority of points occur in areas high intensity. The next group of models, DYNAMICO, FVM, and ICON, comprise members that were part of both the high intensity and low intensity model groups in Figure 1. The wind-pressure relationships of these models are similarly non-linear with most of the points occurring in the high intensity region. In this case, the variance among the points is more evenly distributed among the entire range. The final group of models, CSU-CP, CSU-LZ, GEM, and NICAM, are the models that tended to have the lowest intensities and be the most linear, in part due to relatively weak intensities. NICAM has very little variability except in areas of low intensities, which was expected based on the smoothness of the evolution curves in Figure 1.

### 3.3 Horizontal and Vertical TC Structure

The horizontal and vertical structure of the simulated TCs are analyzed using radial profiles of 1 km wind speed and surface pressure (Figure 3) and radial wind composites (Figure 4), all of which were azimuthally averaged. The wind speed rapidly increases until it reaches its maximum value of 25-50 $\mathrm{ms}^{-1}$ depending on the model. For all models except GEM and NICAM, this maximum occurs at an approximate radius of 100 km. The maximum wind speed for GEM occurs slightly closer to the TC center while NICAM's maximum wind speed occurs at a radius of approximately 200 km. After the radius of maximum wind speed the wind speed decreases exponentially in all models, slowly approaching 0 $\mathrm{ms}^{-1}$ at large radii. All models have wind speeds below 10 $\mathrm{ms}^{-1}$ at radii greater than 600 km, and likely linked to the identical physical environment TCs are initialized in. The results shown here are similar to theoretical and observed azimuthally averaged surface wind radial profiles given in





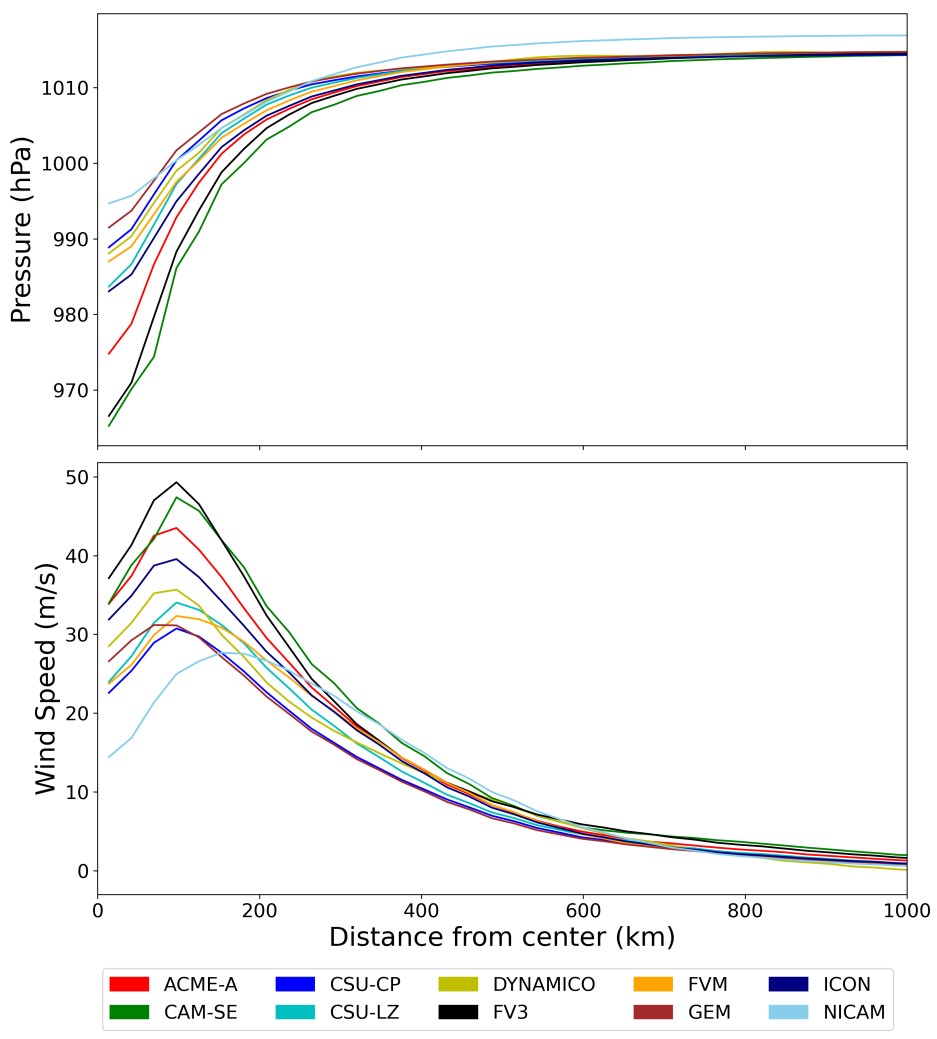

**Figure 3.** Radial profiles of 1 km wind and surface pressure averaged from days 4-10 of the 50 km simulation. Values in the radial profiles are azimuthally averaged.





Chavas and Lin (2015) and Chavas et al. (2017). While the observed radial profiles in Chavas and Lin (2015) tend to have smaller radii of maximum winds, the general structure of the radial profile is in agreement.

In all models, the MSP value occurs at the storm center, as expected, and ranges from approximately 965 hPa to 995 hPa. The most intense models with the lowest minimum pressures are ACME-A, CAM-SE, and FV$^3$ which have minimums in the 965-975 hPa range, while all other models have minimums in the 985-995 hPa range. The pressure values in all models then

rapidly increase until an approximate radius of 200 km, after which they continue to increase at a slower rate until around a 400 km radius. This is a likely reason for the radius of maximum wind being under 200 km in almost all cases. At greater radii, the pressures remain relatively constant and approach the prescribed surface pressure value of 1013 hPa consistent with Section 2.1.1 and the initialized environment. This behavior occurs in all models except NICAM, which plateaus at a pressure slightly above 1013 hPa, possibly due to an initialization error. The radial surface pressure profiles in Chavas et al. (2017) have very

similar general radial structure for more realistic GCM simulations.

The simulated TCs all have similar vertical composites. Starting from the TC center, the azimuthally averaged wind speeds are very low (as expected) and then quickly increase with radius. The region of most intense TC winds is usually centered around a 100 km radius where they tend to reach a maximum altitude of around 5-10 km, depending on the model, and a maximum width in the radial direction of 100-200 km. There are some exceptions to this as NICAM's wind field has a flat top

and no significant peak while FV$^3$ has a maximum altitude above 10 km. The most intense winds occur in ACME-A, CAM-SE, FV$^3$ and to a lesser extent ICON, where wind speeds are greater than 40 ms$^{-1}$ compared to 30-35 ms$^{-1}$ for the other models. Similar results are seen in more comprehensive GCM simulations analyzed in Moon et al. (2020), where the wind fields of simulated TCs in GCMs have similar structure, specifically with respect to the 2D shape of the most intense winds.

Based on the structural properties of the wind composites, the models can be placed into similar groupings seen in Figures

1, 2, and 3. The most intense models are again ACME-A, CAM-SE, and FV$^3$. DYNAMICO and ICON have regions of strong winds; however, they did not form at the same altitude, width, shape, or intensity as the previous three models. CSU-CP, CSU-LZ, FVM, and GEM only show small signs of intense wind formation, and are largely unable to simulate winds above 35 ms$^{-1}$ in the wind composite. In some cases, the structures present in the intense models are replicated at a weaker scale. Overall, the general structure of the wind composites is similar, with the exception NICAM, indicating that even with the differences, a

variety of GCM dynamical cores produce a TC with consistent characteristics .

### 3.4 Impact of Finer Grid Spacing

The previous analysis is now repeated for participating models (Table 5) that submitted a 25 km horizontal grid spacing simulation. This analysis allows us to document how grid spacing impacts the behavior of an idealized TC in an intermediate complexity environment across GCMs. The time evolution of MSP and MWS is first examined, and results are seen in figure

5. For both MWS and MSP, the evolution largely resembles the coarser grid spacing case from 1 with a period of rapid intensification in the first 4 days followed by an approximate steady state region. In almost all cases, the 25 km simulations are more intense than their 50 km counterparts, and the most intense models in the 50 km simulation are also the most intense in the 25 km simulation. ACME-A and CAM-SE are the most intense models in both grid spacing, with their MWS increasing



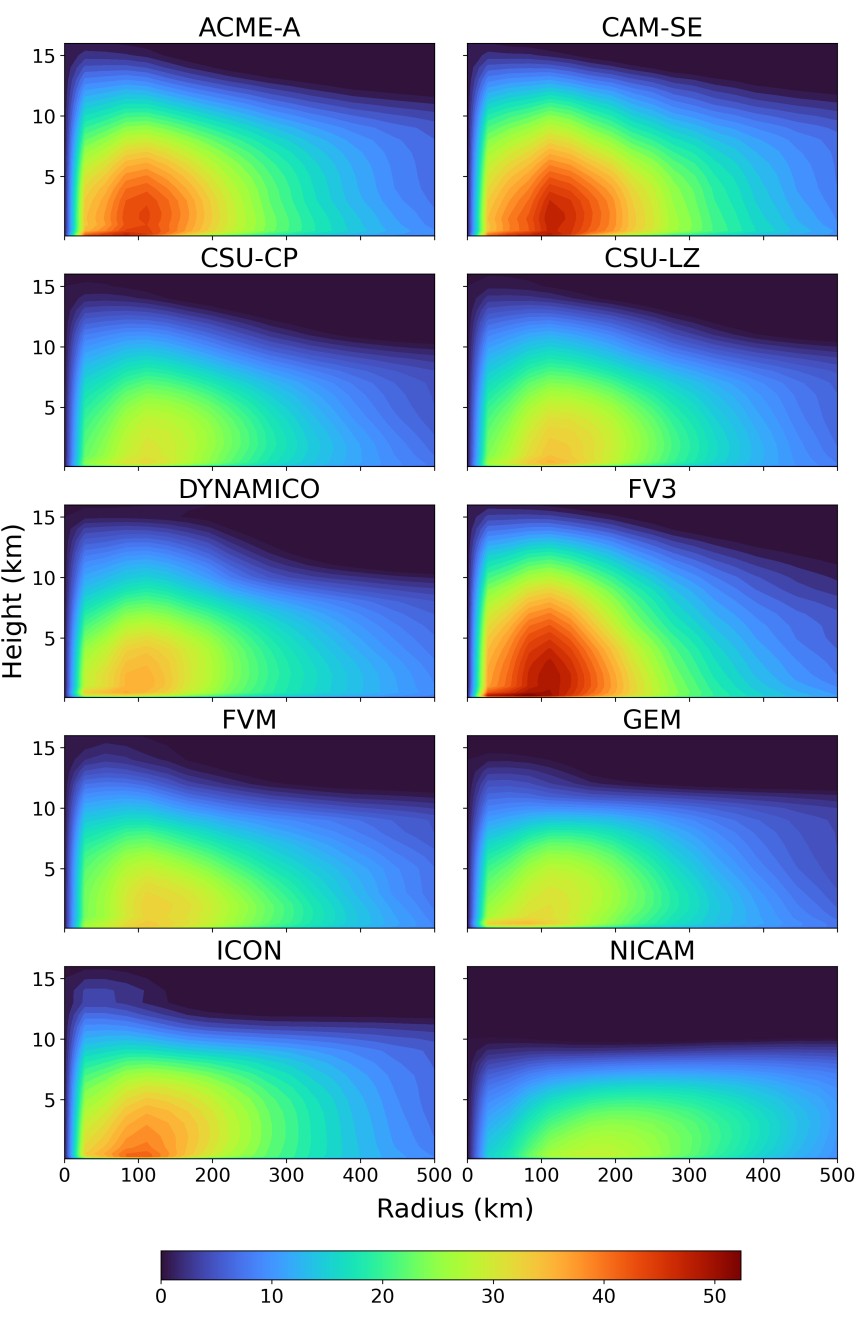

**Figure 4.** Azimuthally averaged vertical wind composite of the simulated TC from days 4-10 of the 50 km simulation.

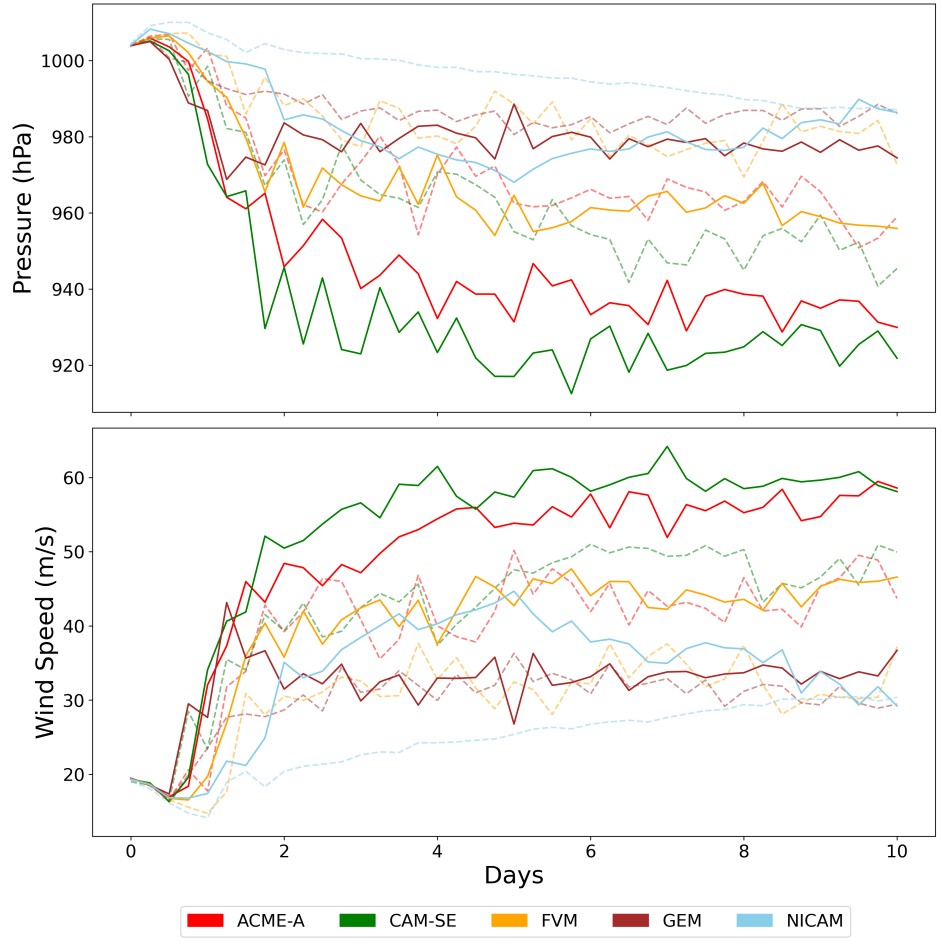

**Figure 5.** Evolution of MSP and MWS over the 10 day simulation period. 50 km (dashed line) and 25 km (solid line) grid spacing are shown for participating models.

from 40-50 ms$^{-1}$ to 50-60 ms$^{-1}$ and their MSP decreasing from approximately 950-970 hPa to 920-940 hPa with decreased
grid spacing. An increase in TC intensity in the CAM model with finer grid spacing is shown in several studies including
Reed and Jablonowski (2011a, c); Reed et al. (2012). This increase in intensity can be related to implicit and explicit diffusion
being weaker at finer grid spacing (Jablonowski and Williamson, 2011). FVM and GEM are again models with intermediate
intensity, and FVM tends to have a larger increase in intensity than GEM by approximately 10 ms$^{-1}$ for MWS and 20 hPa
for MSP compared to under 5 ms$^{-1}$ and around 5 hPa for GEM. NICAM is unique in this analysis because of its substantial
increase in intensity, upwards of 15 ms$^{-1}$ for MWS and 30 hPa for MSP, but these large changes only occurred during days
2-8 of the simulation.

The wind-pressure relationships (Figure 6) have a larger MSP and MWS range for all models, which was expected since
Figure 5 demonstrated higher intensities at 25 km grid spacing. As in the 50 km simulations, most of these relationships are



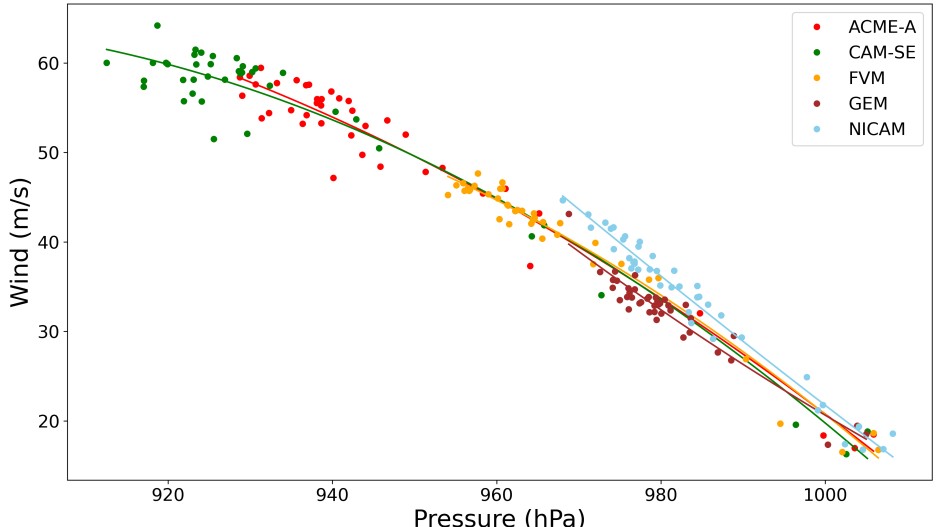

**Figure 6.** Wind-pressure relationship in the simulated TC at all time steps for 25 km simulations of participating models.

non-linear since the rate of increase in MWS tends to decrease at lower MSP. Additionally, a majority of the points occur at the high intensity region as before due to the longer period of the simulation spent by the TC at high intensity. The MSP and MWS values seen in this analysis are within observed ranges for TCs, reaching up to category 4 on the Saffir-Simpson scale.

Radial profiles of 1 km wind speed and surface pressure (Figure 7) are used to determine how TC horizontal structure changes at finer grid spacing. Again, the wind speed increases rapidly with radius until it reaches a maximum, and subsequently decreases exponentially and reaches an asymptotic value of $0 \text{ ms}^{-1}$, as in the coarser grid spacing simulations. At finer grid spacing, this maximum wind speed value occurs at a smaller radius, approximately 50 km compared to 100 km, and has a larger magnitude. All models significantly increase in intensity, often by $10 \text{ ms}^{-1}$ or greater, in the core region.

Surface pressure radial profiles at finer grid spacing also have similarities to those at coarser grid spacing. In both cases, the minimum surface pressure values at the center rapidly increase at relatively small radii, and the rate of increase eventually slows and surface pressure reaches the prescribed value (Section 2.1.1) at large radii. The minimum pressure values decrease in all models by 10-20 hPa. The surface pressure profiles, similar to the wind profiles, are more compact. The 25 km simulations tend to have a more rapid increase that slows at a smaller radius than the 50 km simulations, consistent with the larger magnitude and smaller radius of maximum winds . Results converge at radii greater than 400 km for all models.

As with the previous quantities, grid spacing has an impact on the wind vertical composites, which are shown in figure 8. The overall 2D structure of the 25 km TCs remains similar to that of the 50 km TCs, but there are key differences. As before, there is a narrow region of weak winds by the TC center at all heights followed by a stronger wind field that extends to a radius of approximately 300 km and a height at or above 10 km. In the models that produce the most intense TCs (ACME-A and CAM-SE), there is a region of intense winds that is more compact in the 25 km simulations, extending to around a 100 km radius (versus a 200 km radius in the 50 km simulations), and with stronger winds that are routinely greater than $50 \text{ ms}^{-1}$. This





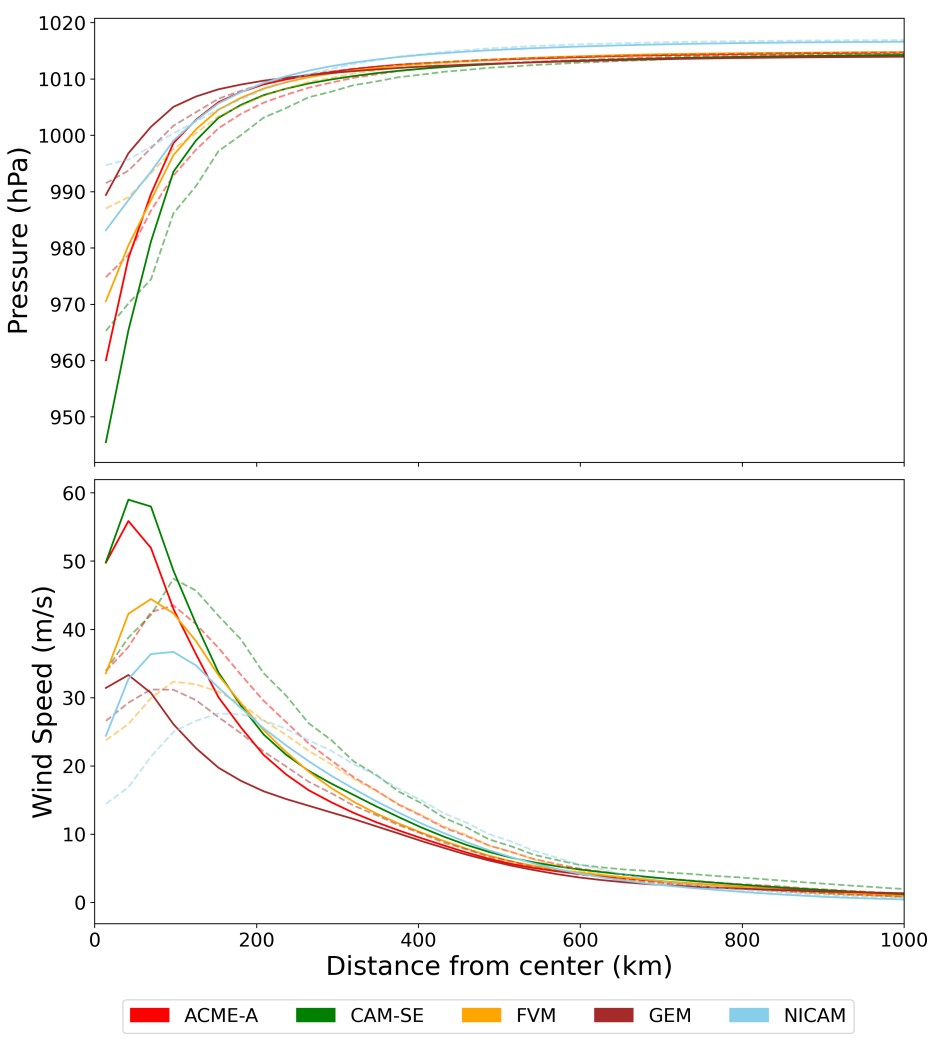

**Figure 7.** Radial 1 km wind and surface pressure profiles averaged from days 4-10 of the simulation. Values in the radial profiles are azimuthally averaged. 50 km (dashed line) and 25 km (solid line) values are shown for participating models.



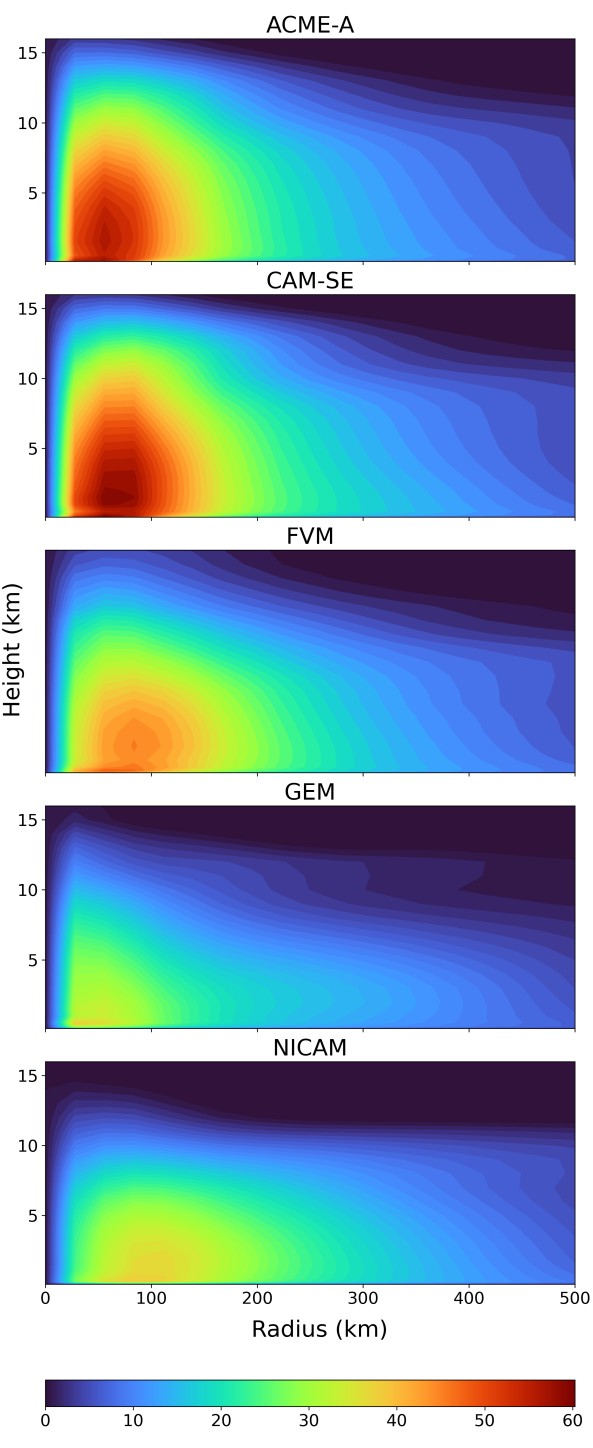

**Figure 8.** Azimuthally average vertical wind composite of the simulated TC from days 4-10 of the 25 km simulation.





compaction in the radius of maximum winds is seen in the remaining models as there is a 50-100 km decrease in GEM, FVM,
and NICAM. GEM, in particular, becomes much more compact, especially at altitudes greater than 5 km, and has a profile with
a different shape overall. This phenomenon also occurs in Moon et al. (2020) when GCM horizontal grid spacing was reduced
in more comprehensive GCM simulations.

## 4  Conclusions

The RJ TC test case results demonstrate that simulations vary between DCMIP2016 models with different dynamical cores
initialized in the same physical environment at the same horizontal grid spacing, building on the work of Reed and Jablonowski
(2012). All participating models produce relatively consistent results; however, there are important differences in the evolution
of MWS and MSP, the wind-pressure relationship, radial profiles of wind and pressure, and wind composites of the TCs.
Certain models tend to be more intense overall, and that is reflected in their MWS, MSP, and structure. These differences are
likely tied to the *effective resolution* of the dynamical core, which is the shortest wavelength which is accurately simulated in
the model (Kent et al., 2014). More intense models also show unique characteristics in their wind composites, such as regions
of intense wind speeds not seen in the other models. TC behavior among participating models also changes when the horizontal
grid spacing becomes finer. TCs simulated at 25 km grid spacing tended to be noticeably more intense and compact than those
simulated at 50 km grid spacing. Models that produced the most intense TCs at 50 km also produce the most intense TCs at 25
km, indicating that some differences between the models are preserved at finer grid spacing.

It is evident that the dynamical core has an essential role in determining the resulting TC behavior in GCMs. While the
impact of the dynamical core has been investigated thoroughly in studies of one or two models, the intercomparison of a larger
group of models illustrates this role and related sensitivity to horizontal grid spacing. The dynamical core choice should be
carefully considered in the GCM development process, and more work can be done to better quantify its effects when all
other parameters are held constant. The goal of this study was to present a general intercomparison of TC behavior among a
grouping of models that differed in dynamical core. In doing so, this work provides a library of solutions that can serve as a
benchmark for modeling groups to compare against during the model development process, similar to other non-TC focused
intercomparison efforts (e.g. Blackburn et al., 2013; Zarzycki et al., 2019). This is especially important since the RJ TC test
case and other DCMIP2016 test cases are widely used in the community and some test cases are readily available in CESM.
Future work could examine differences between specific dynamical core characteristics and how those differences impact TC
simulation in intermediate complexity simulations.

*Code and data availability.* Information on the availability of source code for the models featured in this paper can be found in Ullrich et al.
(2017). For this particular test, the initialization routine, microphysics code, and sample plotting scripts are available at https://doi.org/10.
5281/zenodo.1298671 (Ullrich et al., 2018). Data used in this study will be uploaded to Dryad upon publication.



*Author contributions.* JLW and KAR prepared the text and corresponding figures in this paper. KAR, CJ, JK, PHL, RN, PAU and CMZ led the DCMIP2016 workshop. Data and notations about model-specific configurations were provided by all co-authors representing their modeling groups.

*Competing interests.* The authors declare that they have no conflict of interest.

*Acknowledgements.* DCMIP2016 is sponsored by the National Center for Atmospheric Research Computational Information Systems Laboratory, the Department of Energy Office of Science (award no. DE-SC0016015), the National Science Foundation (award no. 1629819), the National Aeronautics and Space Administration (award no. NNX16AK51G),the National Oceanic and Atmospheric Administration Great Lakes Environmental Research Laboratory (award no. NA12OAR4320071), the Office of Naval Research and CU Boulder Research Computing. This work was made possible with support from our student and postdoctoral participants: Sabina Abba Omar, Scott Bachman, Amanda Back, Tobias Bauer, Vinicius Capistrano, Spencer Clark, Ross Dixon, Christopher Eldred, Robert Fajber, Jared Ferguson, Emily Foshee, Ariane Frassoni, Alexander Goldstein, Jorge Guerra, Chasity Henson, Adam Herrington, Tsung-Lin Hsieh, Dave Lee, Theodore Letcher, Weiwei Li, Laura Mazzaro, Maximo Menchaca, Jonathan Meyer, Farshid Nazari, John O'Brien, Bjarke Tobias Olsen, Hossein Parishani, Charles Pelletier, Thomas Rackow, Kabir Rasouli, Cameron Rencurrel, Koichi Sakaguchi, Gökhan Sever, James Shaw, Konrad Simon, Abhishekh Srivastava, Nicholas Szapiro, Kazushi Takemura, Pushp Raj Tiwari, Chii-Yun Tsai, Richard Urata, Karin van der Wiel, Lei Wang, Eric Wolf, Zheng Wu, Haiyang Yu, Sungduk Yu and Jiawei Zhuang. We would also like to thank Rich Loft, Cecilia Banner, Kathryn Peczkowicz and Rory Kelly (NCAR), Perla Dinger, Carmen Ho, and Gina Skyberg (UC Davis) and Kristi Hansen (University of Michigan) for administrative support during the workshop and summer school.





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
