# Peer review of "DCMIP2016: the tropical cyclone test case"

_Geoscientific Model Development, 2023_

## Referee Comment (RC1)

**gmd-2023-87 : DCMIP2016: the tropical cyclone test case**

Justin L. Willson et al.

**1 General comments**

This paper summarizes some aspects of the results obtained with the academic tropical cyclone test case that was part of the intercomparison project DCMIP2016. The purpose of the publication is to document the results of a set of 9 GCMs which have been selected from among the project participants in order to create a bench-mark for the RJ TC test case that is frequently used by the GCM community to validate model developments.

It is clearly said in the introduction that it is not the purpose of this article to explain the reasons for the differences between the models. In its current state, the article is very descriptive indeed, so descriptive that the scientific level of the paper is not very high for a scientific publication (it looks more like a student report, something like that).

I understand that the purpose of the paper is not a deep analysis of the reasons behind these differences. But a few more basic diagnostics may help to make the article more interesting. For example, the authors suggest the effective resolution as one of the reason for the large differences. I would then suggest to compute global spectra that would help to compare the effective resolution of the different GCMs.

I am also wondering why there are no information about the trajectory of the TC in the participating models. Even if the trajectories are so close that a figure is not necessary, it would be interesting to know about the TC trajectory in the different GCMs.

The difference between the models in this intercomparison is very large (about 50 hPa for MSP, 30 m/s for MWS in Fig.1...). The discrepancy between the results may call into question the utility of these results as a bench mark. Anyway, I would suggest to discuss more openly the implications of such large differences for the TC intensities in the conclusion.

My opinion is that in its current state this article is too "light" to be published as a scientific paper. At the very least, I'd advise adding a few more advanced diagnostics and further interpretation and analysis of the implications of such large differences between the models for a moist test case where "only" the dyncores are different (same initial condition, same horizontal and vertical resolution, same physics package). Also, the style shows a clear lack of experience in writing scientific papers, so it should be carefully checked in a potential revised version.

**2 Specific comments**

- introduction: it could be useful to say somewhere in the introduction that there is no "truth" for this test case. And then discuss a bit more how such a case can be used by model developers. For example, does the intercomparison gives an ensemble which could be used as an ensemble forecast, i.e. it describes some kind of PDF of the possible TC forecasts.

- l.6 and l.88: "1 km azimuthally average wind speed" : I guess it is 1 km above the surface, is it? (I don't think it is a standard diagnostic, so it should be stated clearly). Also, why

not use the usual 10m wind as in the IBTRACKS data base?

- l.10 : it is not said in the abstract that all GCMs use the same physics package, so it seems strange to conclude about the dynamical cores only. I think it would be important to already mention in the abstract that all models use the same physics parametrizations.

- I suggest to move tables 1, 2 and 3 in an appendix.

- l.99 : it is important to say that it is **the same** simplified physical parametrization package. TC simulations are even more sensitive to the choice of parametrizations than they are to dynamical cores so, in order for the comparison to conclude about the sensitivity with respect to dynamical cores, the physics package must be the same.

- l.175 versus FIG. 1 : there are 9 models, but 10 curves on Fig. 1. What is the difference between CSU-CP and CSU-LZ?

- l.185 : "Simulations were performed ... on an interpolated lat-lon grid" ? I don't think the models use a lat-lon grid for their computation, and anyway, it is in contradiction with the next sentence and table 3. I would also remove the "identical" in the next sentence as they are clearly not identical (maybe the protocol is, but the runs are not).

- l. 196-208 : move the reference to the physics package to the beginning of the paragraph and then give the main characteristics of this simple parametrization package. Also, comment about the lack of deep convection scheme in this package as, for models using grid space of 50-25 km, the convection scheme is essential to get realistic moist processes, especially in the tropics.

- section 2.3 : I suggest to move the commands used for the software "TempestExtremes" in appendix. A more detailed description of how the tracking algorithm works would however be useful.

- l. 273 :"rapid intensification" : there is an "official" definition of rapid intensification (more than +30 kt in 24h). Is it what you mean?

- l.312 : it may be exactly the definition of the center (it depends of the detail of the tracking algorithm, so please give more details in section 2.3) so not a very useful comment if is it actually the case...

- l.319 : Should NICAM really stay in the intercomparison? It should be possible to verify if there is, or not, a problem in the initialization/initial conditions of this model. If there is, NICAM should be removed from the intercomparison. But maybe there isn't and there is an extreme numerical diffusion in this model which is smoothing a lot the TC intensification and change the total mass of the model (that would also explain the pressure increase far from the TC centre). A simple plot of the fields at t=0 and a KE spectra later in the forecast should help decide what's going on with NICAM (and if it is worth keeping it in the intercomparison).

- l.316 : what is exactly the "likely reason", and why only likely?

- l.319 : what else would you expect for an azymutally average mslp or low level wind profile around a TC center? Seems to be a very basic characteristics of a TC. Not sure it needs a reference.

- l. 334-335 : I don't see the point of this sentence here. Even real TCs "look" like that. Maybe this sentence should be moved at the beginning of Fig. 4 description to say that the general shape of the mean vertical wind structure in all models is consistent with what is expected in a TC, and then explain the specific differences between the models.

- l. 384 : If you could "prove" that this is the main reason for these very large differences, it would add a lot of value to the paper. But there may also be other reasons for the differences, such as, for example, the way the physics and dynamics are coupled (where and how the physics is called in the time step, etc). These other reasons could also be discuss in the conclusion.

- conclusion : It would be interesting to discuss the magnitude of the differences with regard to, for example, the mean intensity errors in the current NWP models and the TC intensity trends predicted in the climate change scenarios.

- conclusion : I would also suggest to discuss the fact that this test case doesn't use a convection scheme unlike what is usually done in GCM at this resolution. This may "amplify" the impact of the differences between the dynamical cores in these simulations compared to what it would be in a context where the deep convection is treated by a (common) convection scheme. Not sure, but it could be interesting to discuss this aspect in the conclusion.

**3  More technical/form/writing corrections**

My mother tongue is not English, but I have nevertheless the impression that the level of language of the text is not very good. The authors should check it carefully (words missing, bad choice of words etc).

- l 8. : "results are generally similar between all models" : not sure it is what is shown by Fig. 1. There is quite a difference between a TC with a MWS of 25 m/s or 55 m/s. You need to be more accurate with what is actually similar.

- l. 65-65 : a linear response to what? Not clear.

- l.100 : what other characteristics? This sentence (starting l. 98) is not very informative.

- l.123 : not sure "equation" is needed here.

- l. 129 : "perturbation pressure" $\rightarrow$ pressure perturbation

- l.133-135 : this paragraph needs to be reformulated. I don't think "There are multiple pressure differences" means something here. Maybe something like "There are several contributions to the pressure perturbation definition". Then, $r_p$ is not a pressure change, it is a distance, same for $z_p$. These parameters are characteristic scales used to define the shape of the pressure perturbation change in the horizontal and the vertical, they are not the pressure change/decay themselves.

- l. 165 : move the reference to equation 18 just after velocity or at the end of the sentence.

- l. 207 : give a reference for "time-splitting method" and/or give a short explanation. What about the physics-dynamics coupling? Is there any information about that for each model?

- l. 243 : "vary randomly around a fixed mean" : I don't think there are any random processes is these models, reformulate.

- l.244-245 : I think something is missing in the sentence (maybe "to document"?)

- l.264-265: Long and complicated but not very clear sentence : do you mean "decreases linearly with a constant trend".

- l.273-274 : I don't understand what is quantified in section 2.3? What do you mean?

- l.274 : suggestion : the MWS in NICAM....

- suggestion for section 3.1 : move the last sentence to the beginning of the section.

- l.292 : what characteristics? Does it refer to the next sentence or the one before?

- l. 294 : areas **of** high intensity

- l.299 : most linear what?

- l.299 : What does variability means here? Is it fluctuation around the fitting curve?

- l.379: "different dynamical core" then add **but the same physics package**

- l.380 : what do you mean by "physical environment"? Is it the environment of the TC? Or the physics parametrization? Not clear...

- l.382 : be more precise about what you mean by "relatively consistent results" and be more specific about what's different and what's similar (NICAM should probably be treated as an outsider and discuss separately, after checking that there is no error in its setup).

---

## Referee Comment (RC2)

"DCMIP2016: the tropical cyclone test case" Manuscript submitted to GMD by Willson et al..

Evaluating the effects of the dynamical cores coupled with ideal physical parameterization suite by using a ideal test case is an effective way in the scope of atmospheric model development. Reed-Jablonowski (RJ) tropical cyclone (TC) test case which was documented in DCMIP2016 that has been making significant contributions to the design of ideal numerical experiments for model dynamical core, is an idealized tool to study the impact of variable resolutions, physical parameterizations, and numerical method on the simulation and representation of tropical cyclone–like vortices in GCM. In the previous work, the impact of the physical parameterization suite like a dilute plume Convective Available Potential Energy (CAPE) calculation of deep convection on the evolution of an idealized tropical cyclone within the National Center for Atmospheric Research's (NCAR) Community Atmosphere Model (CAM) (Reed and Jablonowski, 2011b) and of the initial-data, parameter and structural model uncertainty (Reed and Jablonowski, 2011c) have been explored.

In contrast, this manuscript describes and analyzes a tropical cyclone test case namely RJ-TC by comparing 9 models like ACME-A, CAM-SE, SCU, DYNMICO, $FV^3$, FVM, GEM, ICON and NICAM in which the used numerics include spectral element (SE), finite difference (FD), and finite volume (FV) and the spherical grids cover cubed sphere, geodesic, Octahehral, Yin-Yang and Icosahedral triangular native grids. This is a comprehensive comparison of RJ-TC simulation results in which evolution of minimum surface pressure and maximum 1 km

azimuthally averaged wind speed, the wind-pressure relationship, radial profiles of wind speed and surface pressure, and wind composites and so on are conducted.

However, it should be noted that the resulting TC behaviors in the 9 model dynamical core coupled with the simple physics package are very different, for example, as Fig. 1, the evolution of MSP can be classified as three categories: a group of ACME-A, CAM-SE and FV$^3$, a group of FVM, GEM, CSU-CP/LZ, DYNAMICO and ICON, a special ICON. Similar situations such as azimuthally averaged vertical wind composite of TC happened in quantitative analysis. Unfortunately, the specific reasons for these differences in outputs are not further elaborated in the manuscript. It would be better if the differences of transport scheme, numerical discretization, artificial diffusion etc. in the corresponding dynamical core and nonlinear interaction of TC could be addressed in details.

In a whole, this manuscript gives comprehensive TC behaviors which provide a valuable library of solutions that serve as a benchmark for modeling groups. I recommend publishing this submission in GMD with the following concerns.

1. For completeness, suggest a table list that describes the simple physics package used in the TC test case. Some physical parameterizations could be addressed in the appendix.

2. If possible, give the detailed transformation formulation between pressure-based level and height level.

3. Due to the 9 model of comparison, it is recommended that the color selected for figures be able to make a significant difference. For instance, the dot colors of CSU-LZ and NICAM is very close in Fig. 2 and it is not easy to recognize them.

4. Please check list of symbol in the table 1. For instance, $q_{cl}$ and

$q_{Cl2}$ seem to be redundant. If some symbols are not used in this manuscript, remove them.

5. Please explain the meaning of abbreviation of "CSU-CP" and "CSU-LZ" in Fig. 1.

6. The superscript of the formula of (4) are prone to ambiguity. Please correct it as $(\quad)^{g/(R_d\Gamma)}$.

7. In Line 472, the paper name of citation is not correct. Please correct it.

---

## Author Comment (AC3)

Dear Reviewer,

Thank you for taking the time to review and make comments on the manuscript *DCMIP2016: the tropical cyclone test case*. We have responded to all comments below and modified the manuscript to reflect this.
* * *
**1 General comments**

This paper summarizes some aspects of the results obtained with the academic tropical cyclone test case that was part of the intercomparison project DCMIP2016. The purpose of the publication is to document the results of a set of 9 GCMs which have been selected from among the project participants in order to create a bench-mark for the RJ TC test case that is frequently used by the GCM community to validate model developments.

- We thank the reviewer for the detailed review of the manuscript. Your attention to detail has significantly improved the manuscript. We have responded to the individual points below.

It is clearly said in the introduction that it is not the purpose of this article to explain the reasons for the differences between the models. In its current state, the article is very descriptive indeed, so descriptive that the scientific level of the paper is not very high for a scientific publication (it looks more like a student report, something like that).

- As is customary with GCMs, sets of solutions that serve as a benchmark for the wider model development community foster innovation and improvements to the models themselves. The purpose of this paper (and other DCMIP papers, e.g. Zarzycki et al. 2019) is to provide those solutions, and despite the fact it is descriptive in nature, we anticipate that individual modeling groups will use these solutions to investigate design sensitivities in more detail.

I understand that the purpose of the paper is not a deep analysis of the reasons behind these differences. But a few more basic diagnostics may help to make the article more interesting. For example, the authors suggest the effective resolution as one of the reason for the large differences. I would then suggest to compute global spectra that would help to compare the effective resolution of the different GCMs.

- With the presentation of DCMIP2016 test case solutions, it is expected that modeling groups will conduct detailed research into the reasons behind the uncertainties in the models. We have added additional descriptions about possible uncertainties and their implications and provided motivation for future work in the conclusion.

I am also wondering why there are no information about the trajectory of the TC in the participating models. Even if the trajectories are so close that a figure is not necessary, it would be interesting to know about the TC trajectory in the different GCMs.

- Due to the configuration of the RJ TC test case, information about the intensity and structure of the TC was prioritized over the trajectories. As expected, the TC trajectories were similar overall (there was a small amount of divergence toward the end of the simulation period) and because of this similarity they were not included in the original manuscript. The trajectories are provided in Figure R1 below and additional, but brief, discussion is included in Section 2.3 of the updated manuscript.

[Figure]

Figure R1. TC Trajectories for the 50 km simulations (left) and 25 km simulations (right).

The difference between the models in this intercomparison is very large (about 50 hPa for MSP, 30 m/s for MWS in Fig.1...). The discrepancy between the results may call into question the utility of these results as a bench mark. Anyway, I would suggest to discuss more openly the implications of such large differences for the TC intensities in the conclusion.

- We provided clearer descriptions of the uncertainties seen in the intercomparison to the conclusion. Specifically, we describe that the physics-dynamics coupling is an additional uncertainty and note that GCMs have been known to have a large intensity spread.

My opinion is that in its current state this article is too "light" to be published as a scientific paper. At the very least, I'd advise adding a few more advanced diagnostics and further interpretation and analysis of the implications of such large differences between the models for a moist test case where "only" the dyncores are different (same initial condition, same horizontal and vertical resolution, same physics package). Also, the style shows a clear lack of experience in writing scientific papers, so it should be carefully checked in a potential revised version.

- We again thank the reviewer for their detailed comments. In addressing them, we clearly see improvement in the manuscript. Although we cannot add more advanced diagnostics in this case, in line with other DCMIP2016 test case results, we have clarified the purpose and implications of the solutions and their uncertainties published here in

several areas of the manuscript. We expect further research to be conducted on model design sensitivities by individual modeling groups.

**2 Specific comments**

- introduction: it could be useful to say somewhere in the introduction that there is no "truth" for this test case. And then discuss a bit more how such a case can be used by model developers. For example, does the intercomparison gives an ensemble which could be used as an ensemble forecast, i.e. it describes some kind of PDF of the possible TC forecasts.
  - This information was added to the introduction. The text now reads "There is no ground truth to this test case, and the RJ TC test case and other DCMIP test cases are in wide use among modeling groups. Standardized test suites are essential for model development, and specific use cases for these tests include verifying the performance of dynamical cores in their operational states and providing assessments of convergence at finer grid spacing and uncertainty between solutions of several models."
- l.6 and l.88: "1 km azimuthally average wind speed" : I guess it is 1 km above the surface, is it? (I don't think it is a standard diagnostic, so it should be stated clearly). Also, why not use the usual 10m wind as in the IBTRACKS data base?
  - This is the wind speed measured at 1 km from/above the surface. A clarification was added to the manuscript where MWS was defined. We did not use 10 m wind as is customary in IBTrACS because the lowest level of wind recorded in the simulation was approximately at a 60-70 m height, which is standard for GCMs. Since the models apply the same PBL parameterization with constant eddy diffusivity, the reduction from the free atmosphere to 10 m will be approximately equivalent for all models, although would be derived from the resolved state. Therefore, while we don't expect the relative results would differ significantly for different heights, we choose 1 km for the wind stability analysis to be well within the TC as resolved by the vertical grid specified in this test.
- l.10 : it is not said in the abstract that all GCMs use the same physics package, so it seems strange to conclude about the dynamical cores only. I think it would be important to already mention in the abstract that all models use the same physics parametrizations.
  - We clarified that the GCMs use the same physics parameterization package in the abstract.
- I suggest to move tables 1, 2 and 3 in an appendix.
  - Although the tables are large, the authors believe they are important to documenting the methodology in the manuscript, especially Section 2.1.
- l.99 : it is important to say that it is the same simplified physical parametrization package. TC simulations are even more sensitive to the choice of parametrizations than they are to dynamical cores so, in order for the comparison to conclude about the sensitivity with respect to dynamical cores, the physics package must be the same.

- - This sentence was clarified to state the physical parameterization package was identical across all GCMs.
- l.175 versus FIG. 1 : there are 9 models, but 10 curves on Fig. 1. What is the difference between CSU-CP and CSU-LZ?
  - We added information on the difference between CSU-CP and CSU-LZ to Section 2.2. The text now reads "CSU submitted two versions of their model, CSU-CP and CSU-LZ, which differ in the vertical coordinate. CSU-LZ uses the Lorenz (Lorenz, 1960) staggering of variables in the vertical, with potential temperature and advected scalars co-located with horizontal winds at mid-layer. CSU-CP used the Charney and Phillips (Charney and Phillips, 1953) staggering of variables with potential temperature and advected scalars co-located with the vertical velocity at the layer interfaces."
- l.185 : "Simulations were performed ... on an interpolated lat-lon grid" ? I don't think the models use a lat-lon grid for their computation, and anyway, it is in contradiction with the next sentence and table 3. I would also remove the "identical" in the next sentence as they are clearly not identical (maybe the protocol is, but the runs are not).
  - This section was clarified to reflect these comments. The simulation was performed on the native grid given in Table 5 and results were interpolated into latitude-longitude grid. Data on both grids were submitted to DCMIP2016 but only the interpolated lat-lon grid was used in this analysis since it was included for all models.
- l. 196-208 : move the reference to the physics package to the beginning of the paragraph and then give the main characteristics of this simple parametrization package. Also, comment about the lack of deep convection scheme in this package as, for models using grid space of 50-25 km, the convection scheme is essential to get realistic moist processes, especially in the tropics.
  - The reference was moved to the beginning of the paragraph and clarification that the physics package used was identical across all models was added as well. An explanation of why convection is not included in the physics package is provided at the end of the paragraph. More information can be found in Reed and Jablonowski 2012.
- section 2.3 : I suggest to move the commands used for the software "TempestExtremes" in appendix. A more detailed description of how the tracking algorithm works would however be useful.
  - The authors believe the commands are better suited in the main text. Specifications were added to the description of the tracking algorithm, and readers were pointed to Ullrich et al. 2021 which contains detailed descriptions of the tracking algorithms used by TempestExtremes.
- l. 273 :"rapid intensification" : there is an "official" definition of rapid intensification (more than +30 kt in 24h). Is it what you mean?
  - We were not using the official definition of rapid intensification here. Descriptions of the TC's intensification were updated accordingly.

- l.312 : it may be exactly the definition of the center (it depends of the detail of the tracking algorithm, so please give more details in section 2.3) so not a very useful comment if is it actually the case...
  - It is the definition of the center and this was updated in the manuscript.
- l.319 : Should NICAM really stay in the intercomparison? It should be possible to verify if there is, or not, a problem in the initialization/initial conditions of this model. If there is, NICAM should be removed from the intercomparison. But maybe there isn't and there is an extreme numerical diffusion in this model which is smoothing a lot the TC intensification and change the total mass of the model (that would also explain the pressure increase far from the TC centre). A simple plot of the fields at t=0 and a KE spectra later in the forecast should help decide what's going on with NICAM (and if it is worth keeping it in the intercomparison).
  - The authors believe that since these are the results that NICAM submitted to the official DCMIP database, they should be included to facilitate a full intercomparison. Test cases like this, even if they have potential initialization errors, are important because they help model groups identify opportunities for future development.
- l.316 : what is exactly the "likely reason", and why only likely?
  - In an effort to streamline the analysis, this sentence was removed from the paragraph.
- l.319 : what else would you expect for an azymutally average mslp or low level wind profile around a TC center? Seems to be a very basic characteristics of a TC. Not sure it needs a reference.
  - Although the characteristics are fundamental, it is important to verify that they are indeed seen in the intermediate complexity simulations. We included the reference to point out this similarity to more complex GCM simulations, and the language regarding the reference has been updated accordingly.
- l. 334-335 : I don't see the point of this sentence here. Even real TCs "look" like that. Maybe this sentence should be moved at the beginning of Fig. 4 description to say that the general shape of the mean vertical wind structure in all models is consistent with what is expected in a TC, and then explain the specific differences between the models.
  - The sentence was removed. Specific differences between the models were explained in the previous two paragraphs.
- l. 384 : If you could "prove" that this is the main reason for these very large differences, it would add a lot of value to the paper. But there may also be other reasons for the differences, such as, for example, the way the physics and dynamics are coupled (where and how the physics is called in the time step, etc). These other reasons could also be discuss in the conclusion.
  - The purpose of this paper is to provide a set of solutions to the modeling community that provide a benchmark for future model development, as commonplace for GCMs (e.g., Zarzycki et al. 2019). Unfortunately we cannot definitively prove the reasons for the large differences via the simulations and output we have. Clearer descriptions of uncertainties in the GCMs were added to the conclusion section, specifically describing that the physics-dynamics coupling

is an additional uncertainty. We anticipate that individual modeling groups will explore model design sensitivities in more detail now that the results from DCMIP have been presented.

- conclusion : It would be interesting to discuss the magnitude of the differences with regard to, for example, the mean intensity errors in the current NWP models and the TC intensity trends predicted in the climate change scenarios.
    - A short discussion on NWP and GCM intensity errors was added to the conclusion. The text now reads "Similarly, numerical weather prediction (NWP) models have large TC intensity root-mean-square errors, often on the order of 2.5-8 m/s depending on lead time (Zhang et al., 2023), although they are smaller in magnitude than the intensity spread seen in this study."
- conclusion : I would also suggest to discuss the fact that this test case doesn't use a convection scheme unlike what is usually done in GCM at this resolution. This may "amplify" the impact of the differences between the dynamical cores in these simulations compared to what it would be in a context where the deep convection is treated by a (common) convection scheme.
    - Comments about the lack of deep and shallow convection in the simulations were added in Section 2.2. The text now reads "Certain parameterizations are not included in order to maintain an intermediate complexity scheme. These parameterizations include radiation, which is not the main driver of cyclogenesis in these short simulations, and shallow and deep convection, which are not necessary in this case since large-scale condensation can form the basis for accurate simulation of ideal TCs at fine horizontal resolution (Reed and Jablonowski, 2012)."

**3 More technical/form/writing corrections**

My mother tongue is not English, but I have nevertheless the impression that the level of language of the text is not very good. The authors should check it carefully (words missing, bad choice of words etc).

- l 8. : "results are generally similar between all models" : not sure it is what is shown by Fig. 1. There is quite a difference between a TC with a MWS of 25 m/s or 55 m/s. You need to be more accurate with what is actually similar.
    - This sentence was updated to be more specific. The text now reads "While all TCs undergo a similar evolution process, some reach significantly higher intensities than others, ultimately impacting their horizontal and vertical structure."
- l. 65-65 : a linear response to what? Not clear.
    - This sentence was updated to be more specific. The text now reads "He and Posselt (2015) demonstrates how the parameterized physical processes in cloud formation, convective development, and moist turbulence impact the simulation of TC intensity, precipitation rate, and other characteristics during the evolution of this RJ TC test case in CAM5, with nonlinear relationships occurring between certain parameters and output variables."

- l.100 : what other characteristics? This sentence (starting l. 98) is not very informative.
  - This sentence was updated to be more specific. The text now reads "GCMs with identical simplified physical parameterization packages simulated this test case in a controlled testing environment to allow for the analysis of dynamical core impact on TC structure and intensity (Ullrich et al., 2016)"
- l.123 : not sure "equation" is needed here.
  - "Equation" was removed.
- l. 129 : "perturbation pressure" → pressure perturbation
  - This phrase was changed.
- l.133-135 : this paragraph needs to be reformulated. I don't think "There are multiple pressure differences" means something here. Maybe something like "There are several contributions to the pressure perturbation definition". Then, rp is not a pressure change, it is a distance, same for zp. These parameters are characteristic scales used to define the shape of the pressure perturbation change in the horizontal and the vertical, they are not the pressure change/decay themselves.
  - This paragraph was clarified. It was not our intention to refer to the variables $r_p$ and $z_p$ as pressure changes themselves but rather describe the shape of the pressure perturbation in the horizontal and vertical directions. The text now reads "The parameters $r_p$, the horizontal half-width of the pressure perturbation, and $z_p$, the height related to the vertical decay rate of the pressure perturbation, describe the shape of the pressure perturbation in these directions."
- l. 165 : move the reference to equation 18 just after velocity or at the end of the sentence.
  - The equation reference was moved to the end of the sentence.
- l. 207 : give a reference for "time-splitting method" and/or give a short explanation. What about the physics-dynamics coupling? Is there any information about that for each model?
  - A short description of the time-splitting method was added here, along with information on physics-dynamics coupling processes in the models. A more detailed description on model processes is found in Ullrich et. al. 2017 and the readers were pointed to that article in the Table 5 caption, Section 2.2, and Section 3. The text now reads "Physics-dynamics coupling is dynamical core dependent and can either be process-split, where $T$ and $q$ are values at the current (previous) time level for two (three) level time schemes, or time-split, where these variables are partially updated prior to physical forcings by the dynamical core's time tendencies."
- l. 243 : "vary randomly around a fixed mean" : I don't think there are any random processes is these models, reformulate.
  - This sentence was changed to accurately reflect the model behavior, and the paragraph as a whole was reformulated to be more clear. The text now reads "The TC intensifies quickly during days 1-4 of the simulation period, consistent with previous analysis of the RJ TC test case (Reed and Jablonowski, 2011b, 2012), and its intensity remains relatively constant at subsequent times. Therefore, the steady state period of this simulation was from days 4-10."

- l.244-245 : I think something is missing in the sentence (maybe "to document"?)
  - "Document" was changed to "to document".
- l.264-265: Long and complicated but not very clear sentence : do you mean "decreases linearly with a constant trend".
  - This sentence was clarified. The text now reads "NICAM retains the highest MSP, and decreases linearly throughout the remainder of the simulation."
- l.273-274 : I don't understand what is quantified in section 2.3? What do you mean?
  - The reference to Section 2.3 pointed to a more detailed description of the steady state period and how this period was determined for analysis. The reference to Section 2.3 was removed as it introduced confusion.
- l.274 : suggestion : the MWS in NICAM....
  - This change was incorporated.
- suggestion for section 3.1 : move the last sentence to the beginning of the section.
  - This change was incorporated.
- l.292 : what characteristics? Does it refer to the next sentence or the one before?
  - The characteristics refer to the next sentence. Due to the confusing wording, the two sentences were combined. The text now reads "ACME-A, CAM-SE, and FV^3 all have few points at low intensities, indicating how they quickly intensify in the first 1-2 days of the simulation."
- l. 294 : areas of high intensity
  - This has been fixed.
- l.299 : most linear what?
  - The wording has been clarified to indicate those models have the most linear wind-pressure relationships.
- l.299 : What does variability means here? Is it fluctuation around the fitting curve?
  - It is the fluctuation around the fitting curve. A definition has been added to the beginning of the paragraph. The text now reads "This intensification occurs with low variability, the fluctuation of points around the fitting curve, but high intensity points tend to vary more."
- l.379: "different dynamical core" then add but the same physics package
  - This has been added to the sentence.
- l.380 : what do you mean by "physical environment"? Is it the environment of the TC? Or the physics parametrization? Not clear…
  - The original wording meant both physics parameterization and TC environment. The updates to the sentence based on the previous comment make the meaning more clear. The text now reads "The RJ TC test case results demonstrate that solutions vary between DCMIP2016 models with different dynamical cores and identical simple physics parameterization packages and physical environments, building on the work of (Reed and Jablonowski, 2012)."
- l.382 : be more precise about what you mean by "relatively consistent results" and be more specific about what's different and what's similar (NICAM should probably be treated as an outsider and discuss separately, after checking that there is no error in its setup).

- The wording of this sentence was updated to be more precise. The text now reads "Most participating GCMs produce a TC with similar MWS and MSP evolution, wind-pressure relationship, radial profiles of wind and pressure, and wind composites; however, there were important differences between them. Certain models were more intense overall, and that is reflected in their MWS, MSP, and horizontal and vertical structure." A sentence about NICAM being an outlier in the intercomparison was added to the end of the paragraph since that in itself was an important result.
* * *
We hope that these updates and comments have addressed your concerns about this manuscript.

Sincerely,

Willson and co-authors

---

## Author Comment (AC4)

Dear Reviewer,

Thank you for taking the time to review and make comments on the manuscript *DCMIP2016: the tropical cyclone test case*. We have responded to all comments below and modified the manuscript to reflect this.
* * *
Evaluating the effects of the dynamical cores coupled with ideal physical parameterization suite by using a ideal test case is an effective way in the scope of atmospheric model development. ReedJablonowski (RJ) tropical cyclone (TC) test case which was documented in DCMIP2016 that has been making significant contributions to the design of ideal numerical experiments for model dynamical core, is an idealized tool to study the impact of variable resolutions, physical parameterizations, and numerical method on the simulation and representation of tropical cyclone–like vortices in GCM. In the previous work, the impact of the physical parameterization suite like a dilute plume Convective Available Potential Energy (CAPE) calculation of deep convection on the evolution of an idealized tropical cyclone within the National Center for Atmospheric Research's (NCAR) Community Atmosphere Model (CAM) (Reed and Jablonowski, 2011b) and of the initial-data, parameter and structural model uncertainty (Reed and Jablonowski, 2011c) have been explored.

In contrast, this manuscript describes and analyzes a tropical cyclone test case namely RJ-TC by comparing 9 models like ACME-A, CAM-SE, SCU, DYNMICO, FV3 , FVM, GEM, ICON and NICAM in which the used numerics include spectral element (SE), finite difference (FD), and finite volume (FV) and the spherical grids cover cubed sphere, geodesic, Octahehral, Yin-Yang and Icosahedral triangular native grids. This is a comprehensive comparison of RJ-TC simulation results in which evolution of minimum surface pressure and maximum 1 km azimuthally averaged wind speed, the wind-pressure relationship, radial profiles of wind speed and surface pressure, and wind composites and so on are conducted.

However, it should be noted that the resulting TC behaviors in the 9 model dynamical core coupled with the simple physics package are very different, for example, as Fig. 1, the evolution of MSP can be classified as three categories: a group of ACME-A, CAM-SE and FV3 , a group of FVM, GEM, CSU-CP/LZ, DYNAMICO and ICON, a special ICON. Similar situations such as azimuthally averaged vertical wind composite of TC happened in quantitative analysis. Unfortunately, the specific reasons for these differences in outputs are not further elaborated in the manuscript. It would be better if the differences of transport scheme, numerical discretization, artificial diffusion etc. in the corresponding dynamical core and nonlinear interaction of TC could be addressed in details. In a whole, this manuscript gives comprehensive TC behaviors which provide a valuable library of solutions that serve as a benchmark for modeling groups. I recommend publishing this submission in GMD with the following concerns.

We thank the reviewer for their detailed and constructive feedback on the manuscript, and for appreciating the value of the manuscript. In addition to the responses below, we do note that it is difficult to elucidate the specific reasons for the differences documented in this manuscript.  That being said, the purpose of this paper is to document a set of solutions to the

modeling community that provide a benchmark for future model development, as commonplace for GCMs. In response to your, and reviewer 1's, comments we have provided clearer descriptions of uncertainties in the GCMs were added to the conclusion section, specifically describing that the physics-dynamics coupling is an additional uncertainty. We anticipate that individual modeling groups will explore model design sensitivities in more detail now that the results from DCMIP have been presented.

1. For completeness, suggest a table list that describes the simple physics package used in the TC test case. Some physical parameterizations could be addressed in the appendix.
- The same simple physics parameterization package was used in all models, which was clarified in Section 2.2. The simple physics package is identical to the one described in Reed and Jablonowski 2012, and more information about it can be found in that paper.

2. If possible, give the detailed transformation formulation between pressure-based level and height level.
- Information about these transformations was added to Section 2.2. The text now reads "In the intercomparison, height levels were used for analysis. Pressure-based vertical levels were converted to height levels by first converting to pressure coordinates if the model utilized hybrid coordinates. The pressure at level k, $p_k$, was obtained using the equation $p_k = a_k p_0 + b_k p_s$ where $a_k$ and $b_k$ are conversion constants at level k, $p_0$ is the reference pressure (table 2), and $p_s$ is the surface pressure at every point (k=0). Then, the pressure levels were converted to height levels using the hypsometric equation $h = z_2 - z_1 = R T_v g \ln(p_1/p_2)$ where $z_1$ ($p_1$) and $z_2$ ($p_2$) are height (pressure) values at adjacent levels and $T_v$ is the mean virtual temperature between the two levels, calculated by the equation $T_v = T(1+M_v q)$. All relevant quantities were then interpolated to desired height using linear interpolation."

3. Due to the 9 model of comparison, it is recommended that the color selected for figures be able to make a significant difference. For instance, the dot colors of CSU-LZ and NICAM is very close in Fig. 2 and it is not easy to recognize them.
- The colors for the models have been updated to be more distinguishable. Additionally, certain figure legends have been updated to have larger icons so these colors can more easily be associated with each model.

4. Please check list of symbol in the table 1. For instance, $q_{(cl)}$ and $q_{(cl2)}$ seem to be redundant. If some symbols are not used in this manuscript, remove them.
- Symbols that were not used in the manuscript were removed.

5. Please explain the meaning of abbreviation of "CSU-CP" and "CSU-LZ" in Fig. 1.
- Information about the differences between CCU-CP and CSU-LZ was added in the beginning of Section 2.2. The text now reads "CSU submitted two versions of their model, CSU-CP and CSU-LZ, which differ in the vertical coordinate. CSU-LZ uses the Lorenz (Lorenz, 1960) staggering of variables in the vertical, with potential temperature and advected scalars co-located with horizontal winds at mid-layer. CSU-CP used the Charney and Phillips (Charney and Phillips, 1953) staggering of variables with potential temperature and advected scalars co-located with the vertical velocity at the layer interfaces."

6. The superscript of the formula of (4) are prone to ambiguity. Please correct it as $()^{(g/(R_d \Gamma))}$

- We corrected this to make it consistent with other equations in the manuscript including (5) and (8).

7. In Line 472, the paper name of citation is not correct. Please correct it.

- The paper name has been updated.
* * *
We hope that these updates and comments have addressed your concerns about this manuscript.

Sincerely,

Willson and co-authors